EMBO
Molecular Medicine

# Inducing mitophagy in diabetic platelets protects against severe oxidative stress

Seung Hee Lee[1], Jing Du[1], Jeremiah Stitham[1], Gourg Atteya[1], Suho Lee[2], Yaozu Xiang[1], Dandan Wang[1], Yu Jin[1], Kristen L Leslie[1], Geralyn Spollett[3], Anup Srivastava[4], Praveen Mannam[4], Allison Ostriker[1], Kathleen A Martin[1], Wai Ho Tang[1,5,*] & John Hwa[1,**]

## Abstract

Diabetes mellitus (DM) is a growing international concern. Considerable mortality and morbidity associated with diabetes mellitus arise predominantly from thrombotic cardiovascular events. Oxidative stress-mediated mitochondrial damage contributes significantly to enhanced thrombosis in DM. A basal autophagy process has recently been described as playing an important role in normal platelet activation. We now report a substantial mitophagy induction (above basal autophagy levels) in diabetic platelets, suggesting alternative roles for autophagy in platelet pathology. Using a combination of molecular, biochemical, and imaging studies on human DM platelets, we report that platelet mitophagy induction serves as a platelet protective mechanism that responds to oxidative stress through JNK activation. By removing damaged mitochondria (mitophagy), phosphorylated p53 is reduced, preventing progression to apoptosis, and preserving platelet function. The absence of mitophagy in DM platelets results in failure to protect against oxidative stress, leading to increased thrombosis. Surprisingly, this removal of damaged mitochondria does not require contributions from transcription, as platelets lack a nucleus. The considerable energy and resources expended in "prepackaging" the complex mitophagy machinery in a short-lived normal platelet support a critical role, in anticipation of exposure to oxidative stress.

**Keywords** diabetes mellitus; mitophagy; oxidative stress; platelets
**Subject Categories** Haematology; Metabolism; Vascular Biology & Angiogenesis

## Introduction

Platelets are short-lived (7–10 days) circulating anucleate cytoplasmic fragments (1.5–3 μm) containing critical factors required for regulation of thrombus formation, vascular homeostasis, and immune response (Lindemann *et al*, 2001; Leytin *et al*, 2007; Alexandru *et al*, 2012; Leytin, 2012). Platelets are capable of many fundamental cellular functions despite being anucleate, including *de novo* protein synthesis (Weyrich *et al*, 1998; Pabla *et al*, 1999; Lindemann *et al*, 2001) and programmed cell death (Vanags *et al*, 1997; Mason *et al*, 2007). A basal autophagy process has only recently been described in platelets (Feng *et al*, 2014; Cao *et al*, 2015; Ouseph *et al*, 2015).

Macroautophagy (hereafter referred to as autophagy) is a ubiquitous, evolutionarily conserved, and tightly regulated process in eukaryotic cells serving to degrade cellular components using the lysosomal machinery (Yu *et al*, 2010; Choi *et al*, 2013). Autophagy plays many essential roles in cell growth, development, homeostasis, and recycling of cellular components (Choi *et al*, 2013). Autophagy is capable of targeting invading bacteria, protein aggregates, and organelles such as mitochondria and endoplasmic reticulum (ER) (Hanna *et al*, 2012; Kubli & Gustafsson, 2012; Choi *et al*, 2013). Recent investigations have identified organelle-specific selectivity in the recognition of autophagy substrates including turnover and quality control of mitochondria through the process of mitophagy (Kubli & Gustafsson, 2012). The autophagy process is highly ordered, with initial phagophore formation (nucleation) requiring the assembly of a complex. Subsequent expansion of the membrane is mediated by ubiquitin-like conjugating systems, microtubule-associated protein 1 light chain 3 (LC3), and the autophagy protein system (ATGs). The phagophore expands, completely surrounding its target, followed by fusion with a lysosome, leading to content degradation by lysosomal enzymes (Shintani & Klionsky, 2004; Kubli & Gustafsson, 2012; Choi *et al*, 2013). While the importance of the nucleus in regulating the process of autophagy has recently been highlighted in an elegant

1   Section of Cardiovascular Medicine, Department of Internal Medicine, Yale Cardiovascular Research Center, Yale University School of Medicine, New Haven, CT, USA
2   Departments of Neurology and Neurobiology, Cellular Neuroscience, Neurodegeneration and Repair Program, Departments of Neurology and Neurobiology, Yale University School of Medicine, New Haven, CT, USA
3   Section of Endocrinology & Metabolism, Yale University School of Medicine, New Haven, CT, USA
4   Department of Medicine, Section of Pulmonary, Critical Care and Sleep Medicine, Yale University School of Medicine, New Haven, CT, USA
5   Guangzhou Institute of Pediatrics, Guangzhou Women and Children's Medical Centre, Guangzhou Medical University, Guangzhou, China
    *Corresponding author. Tel: +1 203 737 5583; Fax: +1 203 737 6118; E-mail: waiho.tang@yale.edu
    **Corresponding author. Tel: +1 203 737 5583; Fax: +1 203 737 6118; E-mail: john.hwa@yale.edu

review (Fullgrabe *et al*, 2013), autophagy/mitophagy can also occur in anucleate platelets. The basal platelet autophagy process appears to play an important role in platelet activation (Feng *et al*, 2014; Cao *et al*, 2015; Ouseph *et al*, 2015).

Diabetes mellitus (DM) is a progressive and chronic metabolic disorder characterized by hyperglycemia caused by impaired insulin levels, insulin sensitivity, and/or insulin action. Currently, over 19.7 million adults in the USA have diagnosed DM and an estimated 8.2 million have undiagnosed DM (Go *et al*, 2013a,b). Sixty-five percent of patients with DM will die from thrombotic events including heart attacks and strokes (Ferreiro & Angiolillo, 2011). Platelets play key roles, leading to thrombotic occlusion of major vessels and tissue death. Of great concern is that 10–40% of DM patients are biochemically insensitive to the most commonly used drug to prevent and treat thrombosis (heart attacks and strokes), aspirin (Price & Holman, 2009; Pignone *et al*, 2010a,b; Ferreiro & Angiolillo, 2011). Compounding the aspirin insensitivity, DM platelets are recognized to be hyperactive (Ferreiro & Angiolillo, 2011; Tang *et al*, 2011) arising from the oxidative stress associated with DM (Kaneto *et al*, 2010). New therapies targeting the underlying mechanisms for thrombosis are urgently warranted, particularly in light of the growing prevalence of DM (38.2% of the US adult population has prediabetes with an abnormal fasting glucose) (Go *et al*, 2013a,b).

In examining platelets from patients with diabetes mellitus, we discovered that the autophagy process and specifically mitophagy were highly upregulated. A role for induced platelet mitophagy in a human disease has never before been reported. This could provide important mechanistic insights into other potential roles for mitophagy in health and disease. We set out to discover how mitophagy was being induced in diabetic platelets and why there was such intense upregulation. Clearly, there must be an important reason for such a short-lived anucleate cell to possess all the machinery to undertake such a complex energy-requiring process. We now present the first report that mitophagy is an intrinsic platelet mechanism that protects DM platelets from severe oxidative stress, preventing apoptosis and thrombosis.

# Results

## DM platelet oxidative stress is associated with p53 phosphorylation, mitochondrial dysfunction, and apoptosis

Hyperglycemia (associated with diabetes mellitus—DM) can lead to enhanced oxidative stress, phosphorylation of p53, and mitochondrial dysfunction and damage, resulting in apoptosis (Polyak *et al*, 1997; von Harsdorf *et al*, 1999; Li *et al*, 1999; Tang *et al*, 2014). Diabetic platelets are subject to substantial oxidative stress arising from chronic hyperglycemia, lipids, and many other factors (e.g. inflammation) (Ferreiro *et al*, 2010). Indeed, there is significantly increased ROS associated with DM platelets when compared to healthy controls (Fig 1A). To demonstrate that such increases in ROS are having substantial effects on the DM platelets, we assessed for protein modifications that are directly induced by ROS including 3-nitrotyrosine (3-NT), aldehyde adducts (4-HNE), carbonyl derivatives (dinitrophenyl (DNP)-derivatized carbonyl), and polyubiquitination (Dhiman *et al*, 2012; Silva *et al*, 2015). Analysis of HC (pooled, $n = 4$) vs. DM platelets (pooled, $n = 8$) demonstrated 5.3-fold (3-NT), 1.2-fold (4-HNE), 1.6-fold (DNP derivatized carbonyl), and 1.5-fold (polyubiquitination) increases with DM platelets (Appendix Fig S1). Severe oxidative stress also leads to increased protein phosphorylation of such key proteins as p53, as demonstrated by Western blot analysis of healthy control vs DM patients (Fig 1B ). Confocal microscopy was used to demonstrate and visualize that phosphorylated p53 (normally located in the nucleus) is increased in the anuclear platelet in DM (Fig 1C). The increased phosphorylated p53 and its translocation to mitochondria is an indicator and instigator for mitochondrial damage in DM platelets (Tang *et al*, 2014) and other organs (Tasdemir *et al*, 2008; Hoshino *et al*, 2013). Indicative of mitochondrial dysfunction (or loss of mitochondria), mitochondrial membrane potential is substantially reduced in platelets from patients with DM as demonstrated by MitoTracker fluorescence (Fig 1D) and flow cytometry (Fig 1E—tetramethylrhodamine methyl ester, TMRM fluorescence). Associated with the loss of mitochondrial membrane

**Figure 1.  DM oxidative stress is associated with p53 phosphorylation, mitochondrial dysfunction, and apoptosis.**

A  ROS levels were measured by flow cytometry after staining with specific oxidative stress detection dyes (CellROX green, Molecular Probes, USA) in HC ($n = 4$) and DM ($n = 4$) platelets (*$P = 0.032$ vs. HC).

B  Western blot analysis of ROS downstream signaling molecules p53 and phosphorylated p53 in HC (#1–3) and DM (#1–8) patient platelets. Quantification analysis on HC ($n = 5$) and DM ($n = 18$) individuals (**$P = 0.0025$ vs. HC). GAPDH served as the loading control.

C  Fluorescent immunostaining for phosphorylated p53 in HC and DM patient platelets (arrows). Graph indicates HC and DM platelets positive for phosphorylated p53 signal. The $y$-axis indicates percentage of phosphorylated p53-positive cells from total cells (**$P = 0.0064$ vs. HC). Representative figure from $n = 5$.

D  Mitochondrial membrane potential ($\Delta\Psi m$) was measured by MitoTracker staining (50 nM for 15 min) in HC and DM patient platelets. Graph indicates signal intensity of MitoTracker fluorescence in HC and DM platelets. The $y$-axis indicates fold of signal intensity in HC and DM platelets (**$P = 1.1608E-05$ vs. HC). Signal intensity of each group was converted to fold change compared with HC values. Representative figure from HC ($n = 6$) and DM ($n = 4$).

E  $\Delta\Psi m$ and platelet apoptosis were measured by flow cytometry analysis. $\Delta\Psi m$ was detected using TMRM, and apoptosis was assessed with annexin V (PS externalization). Representative figure from $n = 3$.

F  Western blot analysis of apoptosis-related pp53, p53, cytochrome c, and active caspase-3 in HC and DM platelets. Tubulin was used as the loading control. Representative figure from $n = 3$.

G  Western blot analysis showing release of cytochrome c (Cyto c) into the cytosol in DM. Representative figure from $n = 3$.

H  EM of platelets mitochondria from HC and DM patients demonstrating examples of mitochondrial damage in DM platelets (DM) compared to a typical healthy mitochondrion (HC). Arrows indicate disrupted outer membranes.

I  Quantification of mitochondria in HC ($n = 33$) and DM ($n = 17$) patient platelets. The $y$-axis indicates average number of mitochondria in HC and DM single-platelet EM view (**$P = 5.50811E-05$ vs. HC).

J  Quantification of healthy and damaged mitochondria in HC and DM platelets.

Data information: All data are expressed as mean ± SD.
Source data are available online for this figure.

potential is an increase in annexin V fluorescence (Fig 1E), a marker for platelet apoptosis. Western blot analysis of DM platelets shows other hallmarks of platelet apoptosis including release of mitochondrial cytochrome c and increased active caspase-3 (Fig 1F and G). We

then visualized mitochondrial damage [disrupted internal and outer membranes (Boland *et al*, 2013) (Ding *et al*, 2012)] from randomly selected platelets on electron microscopy (EM) (HC vs. DM). (Fig 1H —HC vs. DM). In DM platelets, mitochondria numbers per platelet

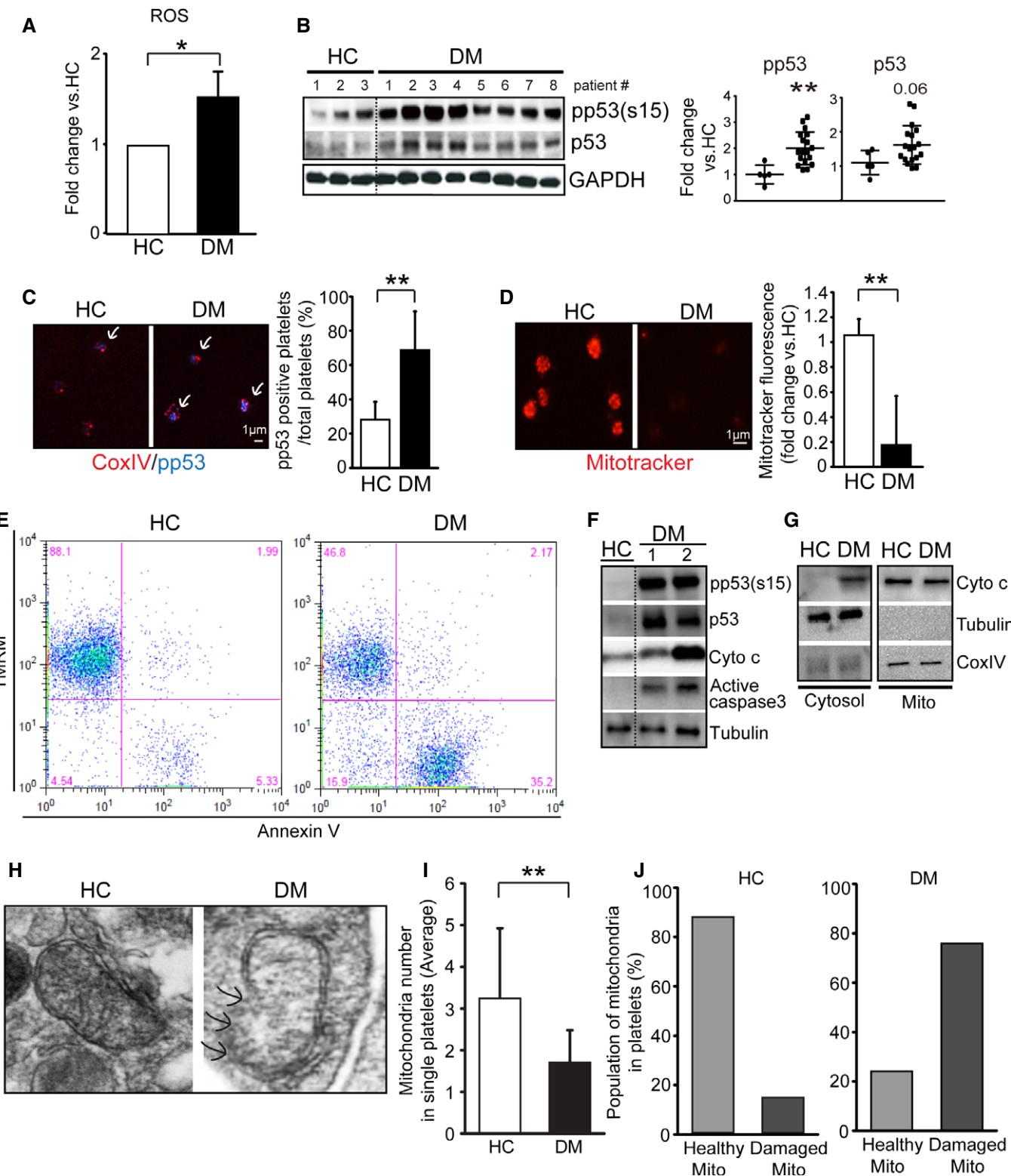

**Figure 1.**

field were reduced by 27% compared with HC (average mitochondria number in HC, $3.2 \pm 1.6$ (100%) $n = 33$; and DM, $1.7 \pm 0.7$ (52.6%) $n = 17$), supporting significant loss of mitochondria (Fig 1I). In DM platelets, damaged mitochondria constituted 75.8% with only 24.1% of mitochondria being normal on ultrastructure compared with 85.0% healthy and 14% damaged in healthy control platelets (Fig 1J). Taken together, these studies support previous studies demonstrating significant oxidative stress in DM platelets leading to increased phosphorylated p53, severe mitochondrial damage, and markers of apoptosis. The next important question to address was whether the anuclear short-lived platelet is able to launch a response to protect itself from such damaging oxidative stress.

## Mitophagy is induced in DM platelets

Recent studies support a basal autophagy process in platelets, important for platelet activation (Feng *et al*, 2014; Cao *et al*, 2015; Ouseph *et al*, 2015). We were intrigued by what appeared to be a distinct induced autophagy process in DM platelets. Autophagy protein components including Beclin1, ATG3, ATG7, ATG12-5, and LC3II were consistently increased in DM patients above healthy control (HC) basal levels (Fig 2A and B). Moreover, electron microscopy examination of healthy control (HC) versus DM platelets demonstrated marked differences in the ultrastructure of HC vs. DM platelets, with substantial vacuolation in the DM platelets (Appendix Fig S2). Detailed examination of DM platelet vacuoles confirmed increased ultrastructural characteristics of autophagosomes (Fig 2C) including early-stage phagophores (Fig 2C DM1), mid-stage autophagosomes (Fig 2C DM2), and late-stage autolysosomes (Fig 2C DM3). To further confirm whether these vacuolated structures were indeed autophagosomes, we performed immunogold labeling (immuno-EM) using the autophagosome marker LC3 (Kabeya *et al*, 2000). In contrast to HC, DM platelets demonstrated increased LC3 (gold particles) which localized and accumulated around what appeared to be mitochondria (Fig 2D).

Confocal microscopy provides a means of further visualizing individual steps in the autophagy/mitophagy process, allowing for examination of LC3 induction and colocalization in individual platelets. This provides support that an active process is taking place in individual platelets. Dissipation of mitochondrial membrane potential (reduced MitoTracker staining as observed with Fig 1D) was associated with increased LC3 (Fig 2E low-powered field, 2F representative single platelet from low-powered field and 2G quantitation—reduced MitoTracker red associated with increased LC3 green in single platelets), suggesting that mitochondrial dysfunction is associated with the induction of mitophagy. In the later stages of autophagy, autophagosomes fuse with lysosomes to generate autolysosomes, leading to lysosomal degradation of autophagosomal components. High-resolution confocal immuno-fluorescence microscopy revealed that LC3 and LAMP1 (lysosomal marker for autolysosomes) were enriched in individual DM platelets relative to healthy controls (Appendix Fig S3A).

Recent investigations have identified organelle-specific selectivity in the recognition of autophagy substrates including turnover and quality control of mitochondria through the process of mitophagy (Kubli & Gustafsson, 2012). As mitochondria function is severely damaged in DM platelets (Fig 1), and based upon our electron microscopy results showing localization of LC3 around

mitochondrial-like structures (Fig 2D), we assessed for the mito-phagy-related proteins PINK1 and Parkin. Parkin is recruited by PINK1 to damaged mitochondria where it ubiquitinates specific substrates responsible for recruiting LC3-conjugated phagophores (Narendra *et al*, 2008; Muller-Rischart *et al*, 2013). Both PINK1 and Parkin were significantly increased in DM platelets (Fig 2A and B). Moreover, Parkin translocated to the mitochondria (Appendix Fig S3B) and colocalized with LC3 on EM (Appendix Fig S3C) and confocal microscopy (Appendix Fig S3D). Appropriate secondary antibody controls showed no significant labeling. BNIP3L/NIX has recently been shown to be involved with mitophagy in reticulocytes (Zhang *et al*, 2012). Increases in BNIP3L/NIX were also observed in the DM platelets further supporting a mitophagy process is present (Appendix Fig S4A and B). We also assessed for mitochondrial DNA content which has been used to assess for reduction of mitochondria (Suliman *et al*, 2007; Lee *et al*, 2011; Dasgupta *et al*, 2015). There was a significant decrease in mitochondrial DNA content in both human and mouse samples (Appendix Fig S5) consistent with reduced mitochondrial numbers observed with Fig 1I. The use of multiple recognized biochemical markers of mitophagy (Feng *et al*, 2014; Cao *et al*, 2015), along with direct visualization in individual platelets, supports an induced mitophagy process in the DM platelet. Taken together, these results (biochemical detection, EM, and immuno-EM) provide direct visualization and detection of increased mitophagy in DM platelets. It is intriguing that such a complex process can be induced in the absence of a nucleus and in a short-lived cell. We postulated that it must serve an important role as substantial energy is required to launch such a complex process.

## Mitophagy is induced in platelets through an oxidative stress-induced JNK pathway

To understand why mitophagy is being induced in the DM platelet, we assessed for recognized autophagy-regulating signals including mTOR/AKT (Shintani & Klionsky, 2004; Kroemer *et al*, 2010) and JNK (Wei *et al*, 2008; Ravikumar *et al*, 2010; Haberzettl & Hill, 2013). Although AKT was of borderline significance, mTOR demonstrated no significant changes (Fig 3A). To confirm that mTOR was not playing a significant role in mitophagy activation, we assessed for pUlk (757), pp70S6K, and pS6 known to be regulated by mTOR (Kim *et al*, 2011). These substrates downstream of mTOR were not significantly increased (Appendix Fig S6A and B). However, JNK phosphorylation (Thr183/Tyr185) was significantly increased (approximately twofold) in the DM platelets ($P = 0.0001$) (Fig 3A). Increased ROS (as with DM cells under hyperglycemia) is recognized to trigger JNK phosphorylation (Wei *et al*, 2008; Kaneto *et al*, 2010; Ravikumar *et al*, 2010). Thus, JNK may play a critical role in the induction of autophagy/mitophagy in DM, possibly linking oxidative stress and mitochondrial dysfunction to induction of mitophagy.

As DM platelets have already been subjected to chronic oxidative stress (Tang *et al*, 2011, 2014), we set out to establish whether we could induce, and control the mitophagy processes, by subjecting healthy platelets to increased levels of oxidative stress observed in DM. It was important to assess healthy platelets as this would demonstrate that mitophagy induction was independent of megakaryocyte (platelet precursor cell) exposure to stresses, with all the mitophagy induction components being prepackaged into a

**Figure 2.  Autophagy is activated in DM platelets.**

A   Western blot analysis of autophagy markers Beclin1, ATG3, ATG7, ATG12-5, and LC3I/II, and the mitophagy-related proteins Parkin and PINK1 in HC (#1–3) and DM platelets (#1–8).

B   Quantification of autophagy markers and mitophagy-related proteins from HC (#1–5) and DM (#1–18) individuals (Beclin1, **$P$ = 0.0003; ATG12, *$P$ = 0.0388; ATG3, *$P$ = 0.0213; ATG7, $P$ = 0.4051; LC3, *$P$ = 0.0298; PINK, *$P$ = 0.0155; Parkin, *$P$ = 0.0103 vs. HC). GAPDH served as a loading control.

C   The three panels represent autophagy stages found in DM platelets. Autophagosome structures containing/enclosing organelles are indicated by the black box. Shown for each panel is an enlargement of the autophagosome (DM1 and 2) or the autolysosome (DM3). Dashed structures are an outline of the autophagosome membranes.

D   LC3 immuno-EM analysis of HC and DM platelets. Arrows indicate immunogold-labeled LC3 clusters. No clusters were found in HC platelets. Representative areas of clusters of gold labeling in DM patients (DM1–3) are presented. Shown in the insets are enlargements of the gold clusters adjacent to mitochondria-like structures.

E   Double staining for MitoTracker (mitochondria membrane potential) (50 nM for 15 min) and LC3 (autophagy) in HC and DM platelets. Boxed platelets are enlarged in (F).

F   A typical HC platelet has high membrane potential and low LC3 in contrast to a representative DM platelet which has low membrane potential and high LC3.

G   Quantification of autophagosome-positive platelets using 6 HC and 4 DM samples after staining using CoxIV (mitochondria) and LC3 (autophagy marker). The $y$-axis indicates percentage of LC3-positive cells in total cells (**$P$ = 0.006 vs. HC).

Data information: All data are expressed as mean ± SD.
Source data are available online for this figure.

healthy platelet in anticipation of oxidative stress. Moreover, induction of mitophagy in healthy platelets exposed to oxidative stress would allow us to dissect details of the relationship between oxidative stress, JNK, and mitophagy in the absence of the complex other metabolic perturbations associated with DM.

Twenty-five millimolar glucose (25 mM high glucose, HG) for 2 h did not significantly induce autophagy (LC3II) in contrast to the addition of $H_2O_2$ which robustly and consistently increased LC3II (Fig 3B). High glucose alone induced only a small increase in ROS (1.5-fold change) versus a threefold change with $H_2O_2$ (Fig 3C). The small increase in ROS with high glucose is indicative of mitochondrial stress, with no substantial mitochondrial damage. The threefold increase in $H_2O_2$-induced ROS is consistent with changes observed in DM platelets (Tang *et al*, 2011, 2014). To demonstrate the importance of $H_2O_2$-mediated ROS, reversal with the antioxidant N-acetyl-cysteine (Zhu *et al*, 2015) was able to reduce autophagosome (LC3II) levels in addition to pJNK and pp53 (markers of severe oxidative stress, Fig 3D). This could also be observed in individual platelets where the highest induction of LC3 (blue) by $H_2O_2$ occurred in platelets with the lowest mitochondrial membrane potential labeling (red) (Fig 3E-insets). Mitochondria (CoxIV—green) that have lost their membrane potential (do not stain with red MitoTracker) colocalize with the autophagy marker LC3 (blue). To further confirm the role of JNK in the ROS-induced enhancement of autophagy, treatment with a JNK inhibitor (SP600125) upon $H_2O_2$ exposure reduced both pJNK and LC3II (Fig 3F). Confirmation was additionally demonstrated in individual platelets where inhibition of JNK reduced LC3II expression (Fig 3G) and decreased LAMP1, as indicated by reduced colocalization of LC3 with LAMP1 (Fig 3G). We also addressed whether p53 itself (increased in DM) may play a role in the induction of platelet autophagy. Inhibition of p53 (with PFT-α) effectively reduced the level of phosphorylated p53, and prevented the mitochondrial membrane potential dissipation, but had no significant effect on LC3II expression (Appendix Fig S7A and B). This was further supported by confocal microscopy analysis of individual platelets (Appendix Fig S7C). These results suggest that while p53 is a critical component for ROS-induced mitochondrial dysfunction, it is not required for ROS-induced mitophagy induction.

Taken together, our data support oxidative stress ($H_2O_2$), leading to JNK activation and LC3II induction. This is strongly supported by chemical inhibition by NAC and the JNK inhibition, in addition to increased oxidative stress, JNK phosphorylation, and LC3II in DM patients. Our results mandate that all the requirements for a robust mitophagy response is present in a healthy platelet, possibly in anticipation of oxidative stress.

### Mitophagy induction protects human platelets from increased oxidative stress

The key question remained as to the role of the induced mitophagy in DM platelets. In the absence of a nucleus and thus transcription, the amount of energy required to prepackage the mitophagy machinery into the platelet is considerable, suggesting that mitophagy plays an important role, possibly in to oxidative stress. Our data thus far support a protective role for mitophagy by removing damaged mitochondria and its associated phosphorylated p53 in DM. Phosphorylated p53 is a key indicator and instigator of both oxidative stress and mitochondria function in DM platelets (Polyak *et al*, 1997; von Harsdorf *et al*, 1999; Li *et al*, 1999; Tang *et al*, 2014). Based upon initial data that $H_2O_2$ (ROS)-induced autophagy through JNK phosphorylation (Fig 3D) and JNK inhibition (SP600125) inhibited autophagosome and autolysosome generation (Fig 3F and G), to assess for function we initially adopted three independent approaches; (i) chemical activation of autophagy (carbonyl cyanide m-chlorophenylhydrazone-CCCP) (Yamagishi *et al*, 2001; Narendra *et al*, 2008; Wang *et al*, 2012), (ii) chemical inhibition of autophagy (3-methyladenine—3MA, and spautin-1), and (iii) genetic autophagy knockdown (ATG3 siRNA) in a human megakaryocyte cell line (MEG-01). Phosphorylation of p53 was significantly attenuated by mitophagy enhancement upon treatment with CCCP (Fig 4A and B), supporting a protective role for mitophagy. Moreover, we used confocal microscopy and colocalization of LC3 and Parkin to demonstrate that CCCP could induce mitophagy in individual platelets (Appendix Fig S8A and B). We then focused on DM platelets where mitophagy is already substantially activated and asked what would happen if we chemically inhibited

---

**Figure 3. Autophagy is activated in platelets through an oxidative stress-induced JNK pathway.**

A   Representative Western blot analysis of recognized autophagy signaling molecules (pJNK, JNK, pAKT, and AKT, and pmTOR, and mTOR) in HC (#1–3) and DM (#1–8) patient platelets. Quantification analysis of autophagy upstream signaling molecules on HC (#1–5) and DM (#1–18) individuals (pAKT/AKT, *P* = 0.0747; pJNK/JNK, **P* = 0.001; pmTOR/mTOR, *P* = 0.3356 vs. HC). GAPDH was used as the loading control.

B   Representative Western blot analysis using HC and HG platelets with or without $H_2O_2$ and assessment for LC3I/II. GAPDH was used as the loading control.

C   Platelet suspensions were incubated in HG for 2 h and with $H_2O_2$ (1 mM). ROS levels were evaluated using a detection kit (Enzo Life Science) for 60 min at 37°C. HG **P* = 0.024 vs. HC group, $H_2O_2$ ***P* = 1.00545E-06 vs. HC group and  $H_2O_2$ ***P* = 0.0008 vs. HG group.

D   Western blot analysis of p53, JNK, and LC3I/II in HC platelets treated with $H_2O_2$ (1 mM for 1 h) alone or with NAC (100 µM for 30 min). GAPDH was used as the loading control.

E   Confocal microscopy was used to corroborate the Western blot analysis using triple staining for MitoTracker, CoxIV, and LC3 in HC or $H_2O_2$. The insets highlight the association of low membrane potential with high levels of LC3 and increased colocalization of Cox IV with LC3 in $H_2O_2$-treated platelets.

F   HC platelets were pretreated with $H_2O_2$ to induce autophagy and then treated with or without SP600125 (JNK inhibitor) at 3 concentrations (1, 5, and 10 µM) to assess for inhibition of autophagy. This Western blot is representative of three independent experiments. Quantification of the three independent experiments is provided on the right using HC, $H_2O_2$ and $H_2O_2$/SP600125(1µM)  results (pJNK/JNK in $H_2O_2$/SP600125, ***P* = 0.0286 vs. $H_2O_2$ group; LC3II in $H_2O_2$, **P* = 0.0241 vs. HC group; LC3II in $H_2O_2$/SP600125, ***P* = 0.0044 vs. $H_2O_2$ group; *n* = 3 for each group).

G   Triple staining of CoxIV, LC3, and LAMP1 in HC and $H_2O_2$ with/without SP600125-treated platelets using confocal microscopy. Arrow indicates localization of LC3 and LAMP1. Graph indicates colocalization between LC3, LAMP1, and CoxIV signal. The *y*-axis indicates fold change of colocalization between LC3 and LAMP1 in the mitochondria area. Signal intensity of each group was converted to fold and compared with HC values ($H_2O_2$, **P* = 0.023 vs. HC; $H_2O_2$/SP600125, **P* = 0.04 vs. $H_2O_2$ group).

Data information: All data are expressed as mean ± SD.
Source data are available online for this figure.

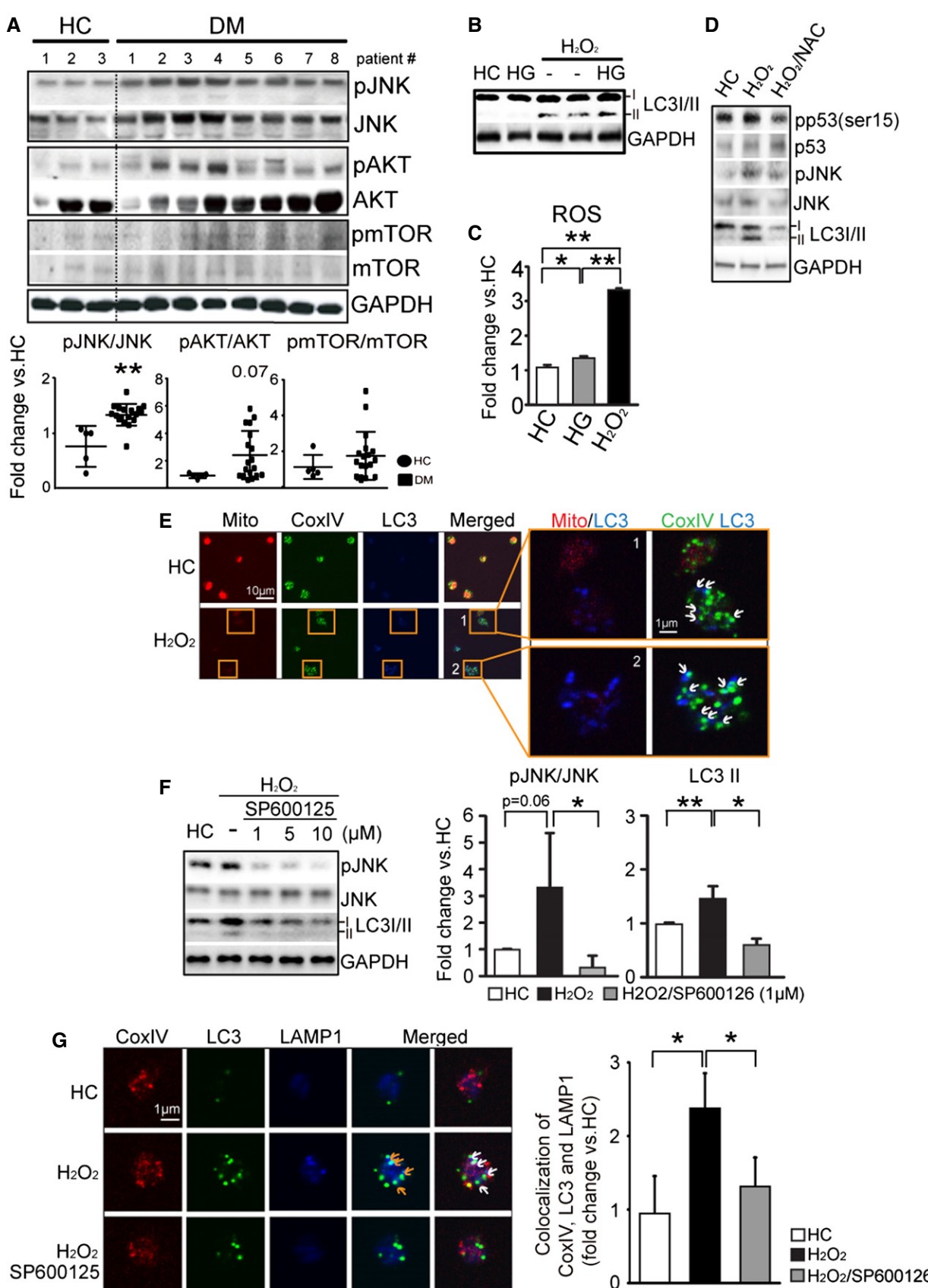

Figure 3.

the process. The addition of the autophagy inhibitor 3MA decreased autophagosome generation (reduced LC3II) and increased phosphorylated p53 significantly, further supporting a protective role for mitophagy (Fig 4C and D), consistent with our JNK inhibition data (Fig 3D). Moreover, the addition of 3MA in the presence of high glucose and $H_2O_2$ substantially increased TMRM-negative platelets and annexin-positive platelets (markers for apoptosis) (Appendix Fig S8C and D). Spautin-1 (another autophagy inhibitor) (Shao *et al*, 2014) also gave a similar result (Appendix Fig S9). To further verify the protective role of autophagy in human platelets, we performed ATG3 siRNA in a human megakaryocyte cell line (Meg-01) and used Meg-01 cell-derived plateletlike particles (PLP) for analysis of phosphorylated p53 upon enhanced oxidative stress. ATG3 regulates conversion of LC3I to LC3II during autophagy (Choi *et al*, 2013). We and others have demonstrated that the platelets derived from Meg-01 cells (PLPs) have similar ultrastructure and

signaling as platelets (Choi *et al*, 1995; Takeuchi *et al*, 1998; Thon *et al*, 2010; Risitano *et al*, 2012). ATG3 expression was substantially reduced in the ATG3 knockdown PLPs and was associated with markedly increased phosphorylated p53 under oxidative stress induced by $H_2O_2$ (Fig 4E and F). Taken together, these data provide support for our hypothesis that mitophagy induction protects platelets from ROS-induced stress. It was previously demonstrated that platelet basal autophagy is critical for normal platelet activation and aggregation (Feng *et al*, 2014; Ouseph *et al*, 2015). Our preliminary studies demonstrated that induction of mitophagy by CCCP significantly reduced platelet aggregation (DMSO = 27.8 ± 12.4%, $n = 4$ and CCCP = 5.1 ± 0.9%, $n = 4$, $P = 0.01$) consistent with mitophagy being protective against mitochondrial (dysfunction and damage)-induced increased platelet activation (thrombosis) (Tang *et al*, 2014; Appendix Fig S10). Removing the dysfunctional mitochondria prevents increased activation and apoptosis, both of which

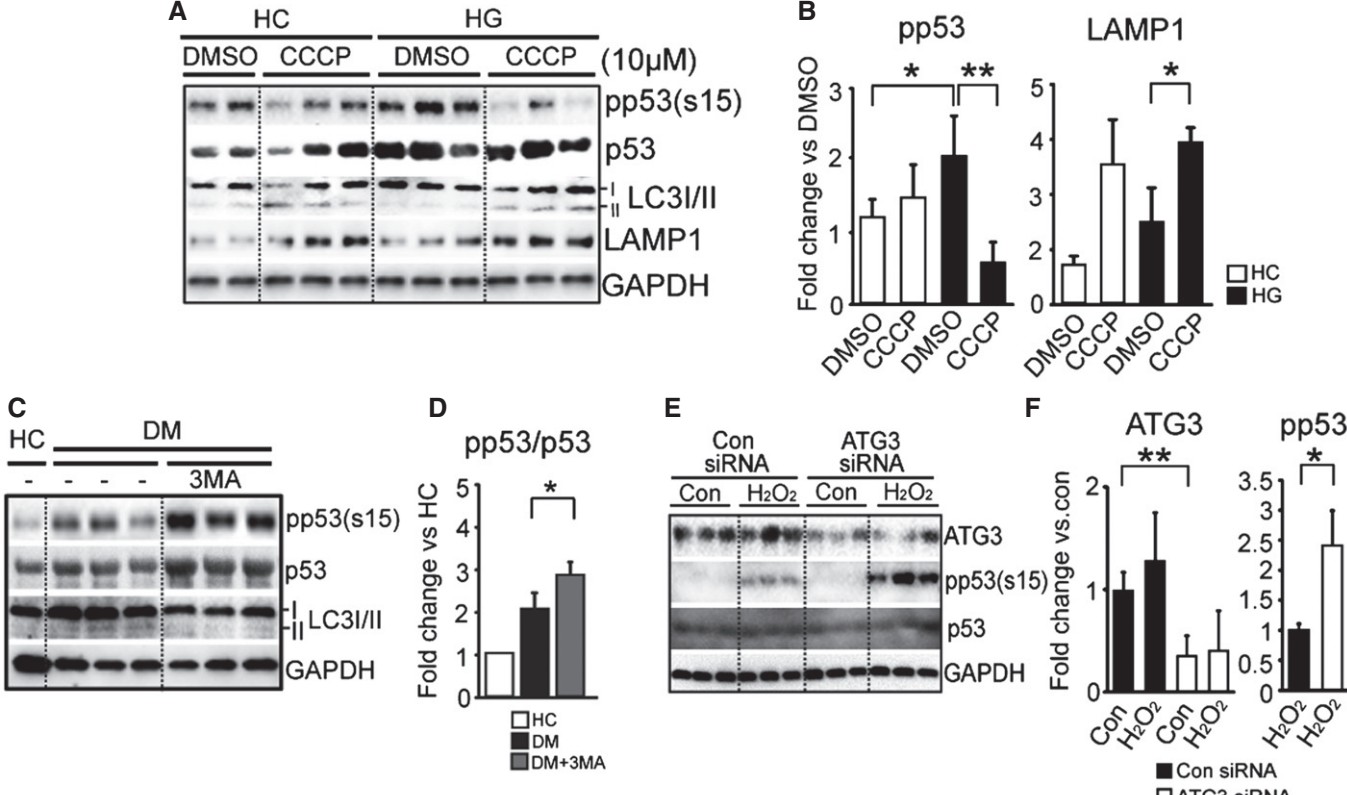

**Figure 4.  Human platelet mitophagy protects platelets from increased oxidative stress.**

A   Western blot analysis of LC3 and the apoptosis markers phosphorylated p53, p53, and LAMP1 in HC platelets. Platelets were treated with high glucose (25 mM) with or without CCCP (10 μM). This Western blot is representative of three independent experiments.

B   Quantification analysis of phosphorylated p53 and LAMP1 (pp53; HG/DMSO, *P = 0.018 vs. DMSO group; HG/CCCP, **P = 0.001 vs. HG/DMSO group; LAMP1; HG/CCCP *P = 0.02 vs. HG/DMSO group; $n = 4$ for each group).

C   The recognized autophagy inhibitor [2 mM 3-methyladenine (3MA) for 1 h] was used to treat DM platelets compared to HC assessing for phosphorylated p53, p53, and LC3I/II.

D   Quantification analysis of phosphorylated p53/p53 was performed. Each value indicated the average band density from a total of three independent samples (DM/3MA, *P = 0.026 vs. DM group).

E   Western blot analysis of PLPs transfected with control or ATG3 siRNA, with or without treatment of 1 mM $H_2O_2$. Each lane displayed a different transfected group.

F   Quantification analysis of ATG3, phosphorylated p53 (Ser15), p53, and GAPDH in each group (ATG3 in ATG3 siRNA, **P = 0.0003 vs control group; pp53 in ATG3 siRNA/$H_2O_2$, *P = 0.0145 vs. $H_2O_2$ group).

Data information: All data are expressed as mean ± SD.

   

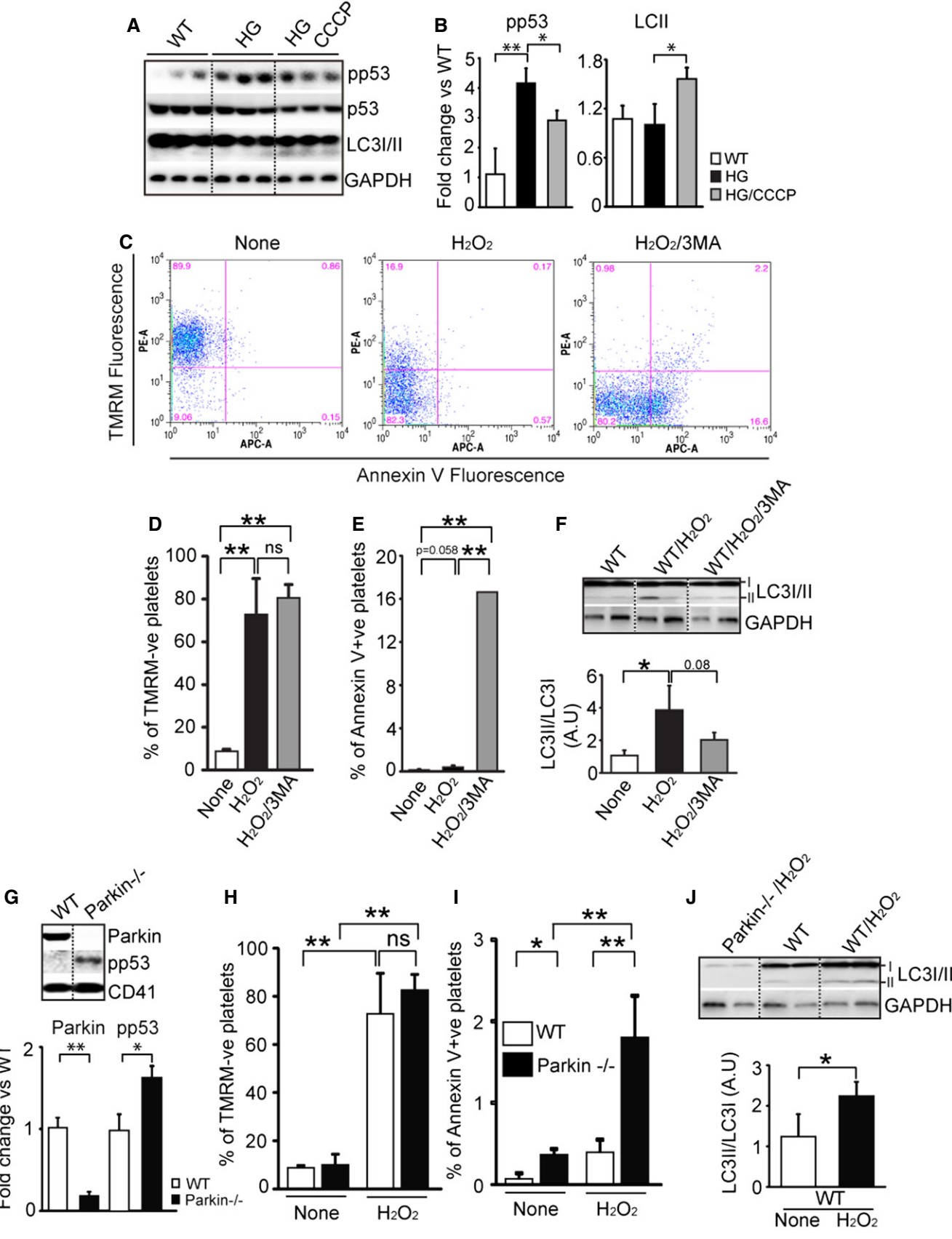

**Figure 5.**

◄

**Figure 5.  Murine platelet mitophagy protects against enhanced oxidative stress.**

A   Western blot analysis of phosphorylated p53 (Ser15), p53, LC3I/II, and GAPDH in WT mouse platelets. Mouse platelets were treated high glucose (25 mM) with or without CCCP (10 μM).

B   Quantification analysis of phosphorylated p53 (Ser15) and LC3 II (pp53 in HG, **$P$ = 0.009 vs. WT group; pp53 in HG/CCCP, *$P$ = 0.022 vs HG group; LC3 II in HG/CCCP, *$P$ = 0.028 vs. HG group; $n$ = 3 for each group).

C   $\Delta\Psi$m and platelet apoptosis were measured by flow cytometry analysis in WT or WT/$H_2O_2$ or WT/ $H_2O_2$/3MA in mouse platelets. $\Delta\Psi$m was detected using TMRM, and apoptosis levels were assessed with annexin V (PS externalization). Representative of $n$ = 3.

D   Graph indicates the percentage of TMRM-positive cells in total cell population ($H_2O_2$, **$P$ = 0.002 vs. not treated group; $H_2O_2$/3MA, **$P$ = 0.008 vs. not treated group; NS means no significance; $n$ = 3).

E   Graph indicates the percentage of annexin V-positive cell in total cell population ($H_2O_2$, NS $P$ = 0.058 vs. not treated group; $H_2O_2$/3MA, **$P$ = 0.004 vs. not treated group, **$P$ = 0.008 vs. $H_2O_2$ group; NS means no significance; $n$ = 3).

F   Representative Western blot analysis of LC3I/II in WT or WT/$H_2O_2$ or WT/$H_2O_2$/3MA mouse platelets (*$P$ = 0.034 vs. WT group, $n$ = 3).

G   Representative Western blot analysis of phosphorylated p53, and Parkin in Parkin$^{-/-}$ mice compared to WT. Quantification analysis of phosphorylated p53 and Parkin (Parkin, **$P$ = 0.0001 vs. WT; pp53, *$P$ = 0.019 vs. WT, $n$ = 4).

H   $\Delta\Psi$m and platelet apoptosis were measured by flow cytometry analysis in WT or WT/$H_2O_2$ and Parkin$^{-/-}$ or Parkin$^{-/-}$/$H_2O_2$ mouse platelets. $\Delta\Psi$m was detected using TMRM (WT/$H_2O_2$, **$P$ = 0.002 vs. WT group; Parkin$^{-/-}$/$H_2O_2$, **$P$ = 1.61525E-06 vs. Parkin$^{-/-}$ group; $n$ = 3)

I   Apoptosis level was assessed with annexin V (PS externalization). Graph indicates the percentage of annexin V-positive cells in total cell population (Parkin$^{-/-}$, *$P$ = 0.010 vs. WT group; Parkin$^{-/-}$/$H_2O_2$, **$P$ = 0.008 vs. WT/$H_2O_2$ group; Parkin$^{-/-}$/$H_2O_2$, **$P$ = 0.001 vs. Parkin$^{-/-}$ group; NS means no significance; $n$ = 3).

J   Western blot analysis in WT and Parkin$^{-/-}$ demonstrating the inability to induce LC3 despite the addition of $H_2O_2$. Shown graphically are the LC3II to LC3I ratios (WT/$H_2O_2$, *$P$ = 0.028 vs. WT group; $n$ = 3).

Data information: All data are expressed as mean ± SD.
Source data are available online for this figure.

---

induce thrombosis (Sinauridze *et al*, 2007; Lannan *et al*, 2014). The study of mitophagy knockout mice was now needed.

## Mitophagy induction protects murine platelets from increased oxidative stress, platelet apoptosis, and thrombosis

As with the human studies high glucose alone increased p53 activation with no significant induction of autophagy in mouse platelets (Fig 5A and B). The addition of CCCP enhanced autophagy and reduced phosphorylated p53 (Fig 5A and B). As further proof for the protective role, we again used the autophagy inhibitor 3MA on murine platelets (Fig 5C–F). After 3MA treatment, almost all of the platelets had lost membrane potential (TMRM) and apoptosis (annexin V signal) was significantly increased compared to $H_2O_2$ treatment alone (Fig 5C–E). We confirmed that treatment with 3MA did indeed lead to reduced LC3II (compared with $H_2O_2$ treatment—Fig 5F). As suggested by the human studies, inhibition of mitophagy leads to substantial mitochondrial dysfunction and progression to apoptosis. To definitively demonstrate that mitophagy is protective, we used a Parkin knockout mouse where mitophagy is genetically inhibited. In the absence of Parkin, the platelets had substantially increased phosphorylated p53 when compared to littermate controls (Fig 5G). Consistent with the human studies, the addition of substantial oxidative stress ($H_2O_2$) led to decreased mitochondrial membrane potential (TMRM signal) and increased apoptosis (annexin V signal), which was further increased in the Parkin$^{-/-}$ mouse platelets (Fig 5H and I and Appendix Fig S11C and D). Figure 5J serves as an important control, confirming the lack of LC3I/II induction (basal levels only) in the Parkin$^{-/-}$ mice even under severe oxidative stress. Significant oxidative stress induces mitophagy, which in turn protects platelets from mitochondrial damage. The absence of mitophagy, either from chemical inhibition or genetic knockout, leads to mitochondrial damage and progression to platelet apoptosis.

To strengthen our findings, we chose to complement the Parkin studies with mice that lacked the other key component of

mitophagy, PINK1 (Zhang *et al*, 2014; Bueno *et al*, 2015). Isolated platelets from PINK1$^{-/-}$ mice, analogous to the Parkin$^{-/-}$ mice, confirmed the presence of only basal autophagy with no induced mitophagy upon $H_2O_2$ treatment (Appendix Fig S11A). There was also increased pp53 also consistent with the PINK1$^{-/-}$ platelets (Appendix Fig S11B). When stressed with severe levels of oxidative stress (as observed with DM), the PINK$^{-/-}$ mice demonstrated substantially increased platelet apoptosis (TMRM-negative and annexin-positive platelets—Appendix Fig S11C, D and E) with increased pp53 (Fig 6A). Platelet apoptosis through production of apoptotic bodies is recognized to increase blood clotting by 50- to 100-fold (Sinauridze *et al*, 2007; Lannan *et al*, 2014). Moreover, the PINK1 knockout has significantly increased in P-selectin (platelet activity) likely due to mitochondrial dysfunction associated with platelet activation (Tang *et al* 2011) (Fig 6B). We were therefore not surprised to observe that the lack of mitophagy (PINK1$^{-/-}$) (in an FeCl₃ thrombosis model) induced very rapid vessel occlusion from thrombosis (Fig 6C and D). The absence of mitophagy under severe oxidative stress thus substantially enhances thrombosis.

## Model of protection from oxidative stress-induced apoptosis and thrombosis by induced mitophagy

Based upon our studies, we developed a model (Fig 7). ROS generated by oxidative stress in diabetes mellitus leads to phosphorylated p53, mitochondria dysfunction, and, if severe, apoptosis (damaging pathway all highlighted in red). Both mitochondrial function and platelet apoptosis are recognized to lead to increased thrombosis. Substantial ROS (as with DM and with $H_2O_2$) induces mitophagy through stress-activated JNK. The mitophagy process (highlighted in blue) removes damaged/dysfunctional mitochondria along with its associated phosphorylated p53, thus protecting the platelet from further severe oxidative stress from the feedback cycle of mitochondria dysfunction and ROS production. Key to the model is the increased ROS. *In vivo* treatment of DM and PINK1$^{-/-}$ DM mice with N-acetyl cysteine reduces ROS (Appendix Fig S12A and D) and

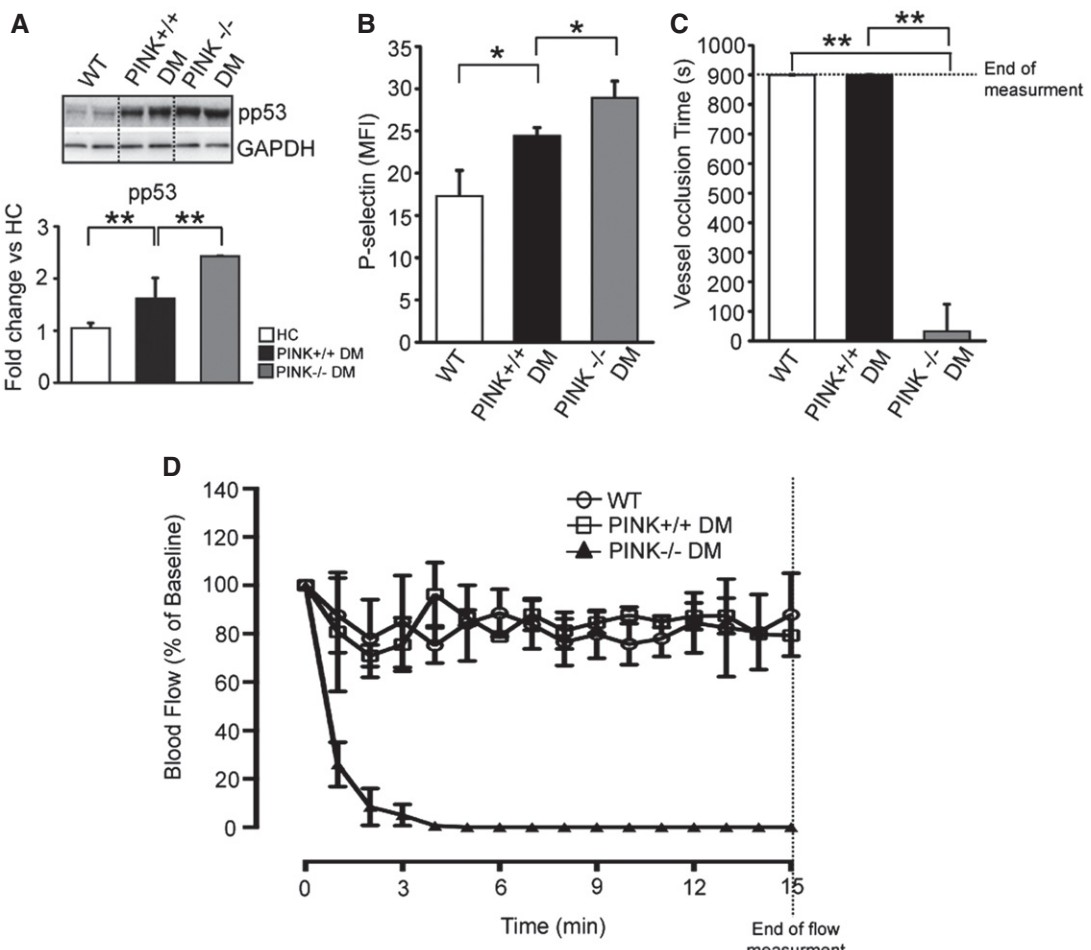

**Figure 6. Activation of platelets is regulated by induced mitophagy.**

A   Representative Western blot analysis of pp53 in WT (*n* = 6), DM (*n* = 6), and PINK1$^{-/-}$ DM mice (*n* = 3). GAPDH served as the loading control (DM, **$P$ = 0.006 vs. WT group; PINK1$^{-/-}$ DM, **$P$ = 0.004 vs. DM group).

B   Measurement of p-selectin translocation to the cell membrane by flow cytometry analysis in WT (*n* = 3), DM (*n* = 3), and PINK$^{-/-}$ DM (*n* = 3) mice platelets. Graphs indicate mean fluorescence intensity (MFI) in each group (*$P$ < 0.05 vs. WT or DM).

C   Quantification of vessel occlusion time after carotid arterial flow measurement (PINK$^{-/-}$, **$P$ < 0.01 vs. WT or DM).

D   Plot of carotid arterial flow by Doppler flow probe after 2 min of FeCl$_3$ injury, indicative of occlusive thrombosis. Flow was plotted as percentage of baseline flow before injury (DM: *n* = 3, PINK$^{-/-}$ DM: *n* = 3).

Data information: All data are expressed as mean ± SD.

thus reduction in platelet apoptosis (Appendix Fig S12B and E). Such reduced ROS also leads to reductions in pJNK, LC3II, and thus mitophagy (Appendix Fig S12C). Thus, the anuclear healthy platelet, despite being short-lived, is able to induce mitophagy to combat anticipated oxidative stress.

## Discussion

The complex, stepwise process of autophagy is an intricate interplay between hundreds of different proteins and lipids leading to digestion of intracellular material. On examining platelets from diabetic patients, we observed that autophagy, specifically mitophagy, is substantially upregulated above basal levels. This raised two important and fundamental questions: Why does a short-lived, anucleate

cell fragment need to contain all the "machinery" to induce such an energy-consuming process, and how is mitophagy induced in the DM platelet? We now present evidence that DM platelet mitophagy protects the DM platelet from oxidative stress-induced apoptosis. Moreover, the normal healthy platelet possesses all the "machinery" to mount a robust mitophagy response, supporting an important role against anticipated oxidative stressors. This induced protection against severe oxidative stress inhibits thrombosis. This role is distinct from basal autophagy which is needed to promote thrombosis. Thus, autophagy has multiple complex roles in the anuclear platelet.

### Protective role of mitophagy in DM platelets

Autophagy has been proposed to be a necessary requirement for degranulation of immune cells, reticulocyte maturation, and

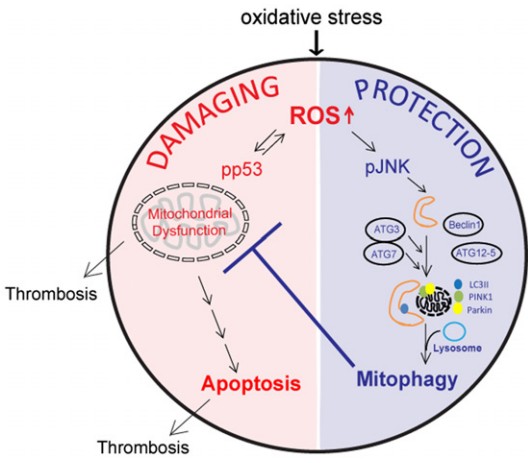

**Figure 7.  Role of mitophagy in DM platelets.**
DM leads to increased oxidative stress and substantially increased ROS. High levels of ROS are recognized to induce phosphorylation of p53 (Ser15). Mitochondrial dysfunction ensues leading to apoptosis and thrombosis (outlined in red). JNK is stress activated (ROS), leading to autophagy enhancement and formation of autophagosomes. Damaged organelles (including mitochondria) are removed through lysosomal fusion and degradation. Mitophagy protects the platelet from oxidative stress by removing the damaged mitochondria and phosphorylated p53 (outlined in blue).

erythroid development (Chen *et al*, 2008; Zhang *et al*, 2009). Despite the absence of nuclei, platelets contain other organelles and specialized structures, including mitochondria and endoplasmic reticulum. As such, platelets are capable of performing a variety of cytoplasmic functions, including *de novo* protein synthesis and translation of proteins (in response to extracellular stress) (Weyrich *et al*, 1998; Pabla *et al*, 1999; Lindemann *et al*, 2001), both of which may contribute to the oxidative stress-activated autophagy process (Kubli & Gustafsson, 2012). In DM, platelets undergo diverse physiological and morphological changes against extracellular stress such as increased plasma glucose, lipid, and ROS (Ferreiro *et al*, 2010), leading to mitochondrial dysfunction and damage (Tang *et al*, 2014). Mitophagy may remove damaged mitochondria and its associated proteins, preventing apoptosis. However, under severe stress, extensive mitochondrial damage may not be overcome by the mitophagy process (Kubli & Gustafsson, 2012; Wang *et al*, 2012; Matsui *et al*, 2013). Moreover, autophagy can also be a double-edged sword with abnormally enhanced autophagy, potentially inducing apoptosis (Shintani & Klionsky, 2004). Thus, the relationship between autophagy and apoptosis is complex and context-dependent. However, in DM platelets it appears that the predominant role for induced mitophagy is protection from severe oxidative stress.

Acute hyperglycemia on healthy platelets induces phosphorylated p53-mediated mild mitochondrial dysfunction and damage, mild ROS increases, and minimal apoptosis, in part due to the protective effects of increased levels of Bcl-$X_L$ in normal platelets as compared to DM (Tang *et al*, 2011, 2014). Substantial oxidative stress such as $H_2O_2$ addition is required for the induction of autophagy in normal healthy platelets. Under chronic severe stress conditions (as with DM), induction of mitophagy is able to protect human platelets from the enhanced oxidative stress, reducing phosphorylated p53 and apoptosis. Phosphorylated p53 is a key indicator of

both oxidative stress and mitochondria function in DM platelets. After phosphorylation, p53 localizes to the mitochondria (Tang *et al*, 2014) and thus removal of damaged mitochondria (mitophagy) also physically removes phosphorylated p53. Additionally, removal of damaged mitochondria would reduce the ensuing oxidative stress, a key player in the phosphorylation of p53 (Tang *et al*, 2014). Taken together, the platelet mitophagy process is carefully orchestrated to protect against mitochondrial damage, ensuing apoptosis and thrombosis. Thus, our mouse and human studies are complementary, supporting mitophagy as protective against severe oxidative stress and thrombosis.

### Regulation of mitophagy in DM platelets

It has been previously recognized that the mTOR/AKT pathway inhibits autophagy (Shintani & Klionsky, 2004; Klionsky, 2007; Choi *et al*, 2013), and interestingly, these pathways also induce platelet aggregation (Yin *et al*, 2008; Aslan *et al*, 2011). In DM platelets, there was no difference in mTOR activity, with AKT activity only increased in the severe DM patients. It is recognized that mTOR/AKT induction of autophagy occurs under conditions of starvation rather than nutrient excess (glucose) (Roberts *et al*, 2014). Consistent with our results, previous studies have demonstrated that JNK can play an important role in autophagy induction (Wei *et al*, 2008; Ravikumar *et al*, 2010; Haberzettl and Hill, 2013;). p53 activity and JNK activity were increased in DM, but autophagosome formation was not increased in high glucose alone as the induction of ROS was modest. Under conditions of high ROS, such as chronic hyperglycemia and acute hyperglycemia with $H_2O_2$ treatment, autophagosomes and autolysosomes were induced by JNK activity. Taken together, there are distinct features associated with the regulation of mitophagy in DM platelets with oxidative stress playing a central role.

Platelet hyperactivity is a hallmark of diabetes, which contributes to the increased thrombotic events, leading to heart attack and stroke (Yamagishi *et al*, 2001; Ferreiro *et al*, 2010). In addition to the protective role of mitophagy in hyperglycemia-induced mitochondrial dysfunction, we also studied the role of mitophagy in thrombosis. We confirmed that induction of mitophagy with CCCP inhibited collagen-induced platelet aggregation in healthy platelets, providing additional links between mitophagy improved platelet function in DM, and reduced platelet aggregation, consistent with previous studies (Terada, 1990; Yamagishi *et al*, 2001). As further support, both Parkin knockout mice and particularly PINK1 knockout mice (lacked mitophagy) had increased oxidative stress, mitochondria dysfunction, apoptosis, and thrombosis. Taken together, our studies demonstrate an important novel role for the induction of mitophagy in an anuclear cell, protection from oxidative stress, and thrombosis.

New therapies targeting the underlying mechanism for the platelet dysfunction are urgently warranted, particularly in light of the growing prevalence of DM (38.2% of the US adult population has prediabetes with an abnormal fasting glucose). In the current study, we demonstrate for the first time that mitophagy can be induced in the DM platelet by oxidative stress leading to protection from organelle damage. The induction of mitophagy may serve as a novel target in treating platelet abnormalities in DM.

    

# Materials and Methods

### Preparation of human platelets

Venous blood was drawn from healthy control and DM volunteers at Yale University School of Medicine (HIC#1005006865). All healthy subjects were free from medication or diseases known to interfere with platelet function (Appendix Table S1). Upon informed consent, a venous blood sample (approximately 20 cc) was obtained by standard venipuncture and collected into tubes containing 3.8% trisodium citrate (w/v). Blood samples were prepared as previously described (Saxena *et al*, 1989). Platelet-rich plasma (PRP) was obtained by centrifugation of blood at 250 $g$ at 25°C for 15 minutes. Platelet counts in the PRP were estimated using an automated cell counter. Platelet-poor plasma (PPP) was obtained by centrifugation of the rest of the blood at 1,400 $g$ at 25°C for 10 min. The PRP was adjusted with PPP to 2–3 × 10^8 platelets/ml suspension. For the washed platelets, PRP was obtained by centrifugation at 250 $g$ for 15 min, and platelets were sedimented at 1,000 $g$ for 15 min and then resuspended in washing buffer (103 mmol/l NaCl, 5 mmol/l KCl, 1 mmol/l MgCl$_2$, 5 mmol/l glucose, 36 mmol/l citric acid, pH 6.5) containing 3.5 mg/ml BSA (Sigma-Aldrich). After sedimentation, the platelets were washed twice in this buffer and resuspended at 2–3 × 10^8 platelets/ml in the buffer composed of 5 mmol/l HEPES, 137 mmol/l NaCl, 2 mmol/l KCl, 1 mmol/l MgCl$_2$, 12 mmol/l NaHCO$_3$, 0.3 mmol/l NaH$_2$PO$_4$, and 5.5 mmol/l glucose, pH 7.4, containing 3.5 mg/ml BSA. Purity of platelet preparation was determined by Western blot analysis using platelet markers (CD41), monocyte markers (CD14), and red blood cell markers (CD235a) (Appendix Fig S13).

### Measurement of mitochondrial membrane potential (ΔΨm), phosphatidylserine externalization through flow cytometry analysis

To assess the ΔΨm and PS externalization simultaneously, platelet suspensions (5 × 10^6 platelets/ml) were incubated with 1 μM tetramethylrhodamine methyl ester (TMRM) at 37°C for 15 mins, followed by staining with 1 μM annexin V at room temperature for 15 mins. The fluorescent intensity was detected by flow cytometry (LSRII). Platelets were identified and gated by their characteristic forward and side scatter properties. A total of 10,000 platelets were analyzed from each sample.

### Immunocytochemistry and confocal microscopy

Washed platelets (1 × 10^6 cells/ml) were allowed to settle on glass-bottomed dishes for 1 h prior to fixing with 4% paraformaldehyde solution (Santa cruz Biotechnology). The platelets were then washed 2 × 5 min in PBS and permeabilized for 5 min in 0.25% Triton X-100/PBS. They were blocked for 60 min in 10% bovine serum albumin (BSA)/PBS at 37°C and incubated in 3% BSA/PBS/ primary antibody for 2 h at 37°C, or overnight at 4°C, and then washed 6 × 2 min in PBS, followed by an additional incubation for 45 min at 37°C in secondary antibody/3% BSA/PBS. Antibodies for LC3 (Cosmo bio, Japan), Parkin (Abcam, USA), LAMP1 and pp53 (Ser15) (Cell Signal, USA), and CoxIV (Santa Cruz, USA) were used.

The stained platelets were observed using a Nikon Eclipse-Ti confocal microscope with 100× oil immersion lens.

### MitoTracker staining

Washed platelets (1 × 10^6 cells/ml) were allowed to settle on glass-bottomed dishes for 1 h and stained mitochondria using MitoTracker (Molecular Probes, MitoTracker Red M7512, or deep Red FM, M22426) for 15 min prior to fixing with 4% paraformaldehyde solution (Santa cruz Biotechnology). The platelets were then washed 2 × 5 min in PBS and permeabilized for 5 min in 0.25% Triton X-100/PBS. They were blocked for 60 min in 10% bovine serum albumin (BSA)/PBS at 37°C and incubated in 3% BSA/PBS/ primary antibody for 2 h at 37°C, or overnight at 4°C, and then washed 6 × 2 min in PBS, followed by an additional incubation for 45 min at 37°C in secondary antibody/3% BSA/PBS. Antibodies for LC3 (Cosmo bio, Japan) were used. The stained platelets were observed using a Nikon Eclipse-Ti confocal microscope with 100× oil immersion lens. Five sets of images were obtained for each condition with a total of 50–100 cells assessed per condition.

### Image analysis

The mean MitoTracker intensity was calculated by Volocity software (PerkinElmer, USA) for healthy and diabetic platelets. The mean intensity was compared between HC and DM and displayed as fold change in a bar graph. Colocalization between LC3 and Parkin or LC3 and LAMP1 was assessed using parameters set in the Volocity software (PerkinElmer, USA). The colocalization was calculated within the red signal area, and the mean colocalization value compared with healthy control or non-treated groups. The data were displayed as fold change.

### Electron microscopy (EM) and immuno-EM

Slides of human platelets were fixed with 2% glutaraldehyde and 2% paraformaldehyde in 0.1 M sodium cacodylate (pH 7.4) for 2 hours at room temperature. They were washed three times with 0.1 M cacodylate buffer at room temperature. Cells were postfixed with 1% osmium tetroxide in 0.1 M cacodylate buffer for 1 h at room temperature. After rinsing with cold distilled water, tissue samples were dehydrated slowly with ethanol and propylene oxide. The samples were embedded in Embed-812 (EMS, USA) and incubated with immunogold-labeled secondary antibody detecting LC3 (Cosmo bio, Japan) and Parkin (Abcam, USA). The samples were visualized using a scanning electron microscope (Yale Biological EM Facility, New Haven, CT). Five sets of images were obtained for each condition. Clustering of immunogold within a distance of approximately 100–200 nm was deemed colocalization with distances greater than 500 nm not localized.

### Western blot analysis

Standard Western blot analysis protocols were used. Thirty micrograms of protein lysates was loaded in each well, and three or more independent replicates were used for quantification. We analyzed the band intensity using ImageJ analysis software (NIH) and

converted the intensity value to fold change in comparison with HC or the non-treated group. Fold values were then used for statistical analysis. Antibodies for Akt, phospho-Akt (Ser473), p53, phospho-p53 (Ser15), JNK, phospho-JNK (Thr183/Tyr185), PINK1, ATG12, ATG3, ATG7, and GAPDH were obtained from Cell Signaling; LC3 from Novus and Abcam, and Parkin from Abcam. A full table of antibodies sources and dilution are provided in Appendix Table S2.

## Measurement of reactive oxidative stress (ROS)

The oxidative stress levels were measured by total ROS detection kit (Enzo LifeScience) and CellROX green (Molecular Probes). Platelets were sedimented by centrifugation at 400 $g$ for 5 min followed by incubation of HC or HG-treated platelets with 1μmol ROS detection mix for 60 min at 37°C (Enzo kit) or HC and DM platelets with 1 μl CellROX green (1:500) for 30 min at 37°C. Changes in fluorescence intensity were measured using a microplate fluorescence reader (BioTak) at excitation/emission wavelengths of 488/520 nm (Enzo kit) or by flow cytometry (LSRII) (CellROX green).

## Plateletlike particle (PLP) isolation from cultured human megakaryocytes (Meg-01 cells)

Meg-01 cells were cultured in RPMI 1640 media (ATCC) with 10% fetal bovine serum. Nucleofection (Lonza) of control siRNA and ATG3 siRNA (300 nM) was performed cells before passage 5. After 48 h of incubation, cells were treated for 5 days with thrombopoietin (TPO) (Takeuchi *et al*, 1998). PLPs derived from Meg-01 cells were purified using a 1.5–3% BSA gradient solution. Isolated PLPs were collected by centrifugation at 300 $g$ for 10 min. Collected PLPs were resuspended in washing buffer (103 mmol/l NaCl, 5 mmol/l KCl, 1 mmol/l MgCl₂, 5 mmol/l glucose, 36 mmol/l citric acid, pH 6.5) containing 3.5 mg/ml BSA (Sigma-Aldrich).

## Preparation of mouse platelets

Blood (0.7–1 ml) was directly aspirated from the right cardiac ventricle into 1.8% sodium citrate (pH 7.4) in WT (C57Bl/6), WT/DM, db/db mice (JAX stock# 000697 and 000642), Parkin$^{-/-}$ (B6.129S4-*Park2*$^{tm1shn}$/J, JAX mice Stock #006582), PINK1$^{-/-}$ (B6;129-Pink1tm1Aub/J, JAX mice stock# 013050) and PINK$^{-/-}$/DM mice (mice were 8 weeks of age; STZ injected for 5 days followed by high-fat diet for 12 weeks). Citrated blood from several mice of identical genotype was pooled and diluted with equal volume of HEPES/Tyrode's buffer. PRP was prepared by centrifugation at 100 $g$ for 10 min and then used for measuring platelet apoptosis and autophagy.

## Measurement of autophagy using flow cytometry analysis

To assess the autophagy activation, platelet suspensions (5 × 10⁶ platelets/ml) were incubated with FITC-conjugated LC3 antibody (Novus, USA) at room temperature for 15 min. The fluorescent intensity was detected by flow cytometry (LSRII). Platelets were identified and gated by their characteristic forward and side scatter properties. A total of 10,000 platelets were analyzed from each sample.

### The paper explained

**Problem**

Considerable thrombovascular mortality and morbidity are associated with the growing population of patients with diabetes mellitus. Platelets play key roles in such increased thrombotic events, in part due to increased oxidative stress leading to mitochondrial damage and apoptosis. Whether the short-lived anuclear platelet can launch a response to protect itself from such insults is not known.

**Results**

We now report a substantial mitophagy induction (above basal autophagy levels) in diabetic platelets that protects the platelet from apoptosis by removing the damaged mitochondria. A combination of molecular, biochemical, and imaging studies was used to demonstrate that platelet mitophagy is induced by JNK activation through oxidative stress. Removal of damaged mitochondria results in reduced phosphorylated p53 and platelet apoptosis, preserving platelet function. The absence of mitophagy in genetic mouse models results in failure to protect against oxidative stress, leading to increased thrombosis.

**Impact**

This new platelet mechanism to protect against the substantial oxidative stress and platelet apoptosis associated with diabetes mellitus may serve as a promising therapeutic target in patients with diabetes mellitus. The need for new adjunct therapy for primary prevention of thrombotic events in diabetic patients is urgently warranted in light of the high prevalence of drug insensitivity to many currently used antithrombotic agents.

## Determination of platelet activation

P-selectin translocation was assessed by flow cytometry analysis using PE-conjugated p-selectin monoclonal antibody (eBioscience). Platelet from WT, DM, and PINK1$^{-/-}$/DM mice (mice were 8 weeks of age; STZ injected for 5 days followed by high-fat diet for 12 weeks) were isolated and fixed in 4% paraformaldehyde solution (Santa cruz Biotechnology) for 15 min. The fixed platelets were washed and labeled with p-selectin antibody for 1 hour. The fluorescent intensity was detected by flow cytometry (LSRII). Platelets were identified and gated by their characteristic forward and side scatter properties. A total of 10,000 platelets were analyzed from each sample.

## Ferric chloride (FeCl₃)-induced carotid artery thrombosis model

For this model, WT, DM, and PINK1$^{-/-}$ DM mice (mice were 8 weeks of age; STZ injected for 5 days followed by high-fat diet for 12 weeks) were anesthetized and target area was externalized for imaging (Zhu *et al*, 2015). One-millimeter rnalize filter paper (Whatman) soaked with 1.5 μl 7.5% FeCl₃ solution was applied to the ventral surface of the carotid for 2 min. To determine the time to occlusion, a Doppler flowprobe (Model 0.5 PSB; Transonic Systems, Ithaca, NY, USA) was placed at the surface of the carotid artery and blood flow was recorded using a Transonic Model TS420 flowmeter (Transonic Systems).

## Mice tail bleeding time

To assess bleeding time, WT, DM, and PINK1$^{-/-}$/DM mice (mice were 8 weeks of age; STZ injected for 5 days followed by high-fat

diet for 12 weeks) were anesthetized before a sharp scalpel cut was made to remove 1 mm of the distal tail. The tail was immersed in PBS at 37°C, and the time required for the blood stream to stop for at least 30 s was recorded in a blinded fashion (Xiang *et al*, 2015).

### Human and mouse studies ethics statement

The animals were housed at the Yale Animal facility 300 George St. New Haven, CT, under the supervision of YARC and Rita Weber (Animal facility manager Yale CVRC). All procedures were approved under animal protocol #11413 (Yale IACUC). C57BL/6 (8 weeks of age) were the background for WT and Parkin or PINK1 knockout mice. We drew blood from the heart after STZ /HFD (50 mg/kg for 5 days followed by high-fat diet for 12 weeks. As per approved protocol, the mice were exsanguinated under deep anesthesia with high-dose ketamine/xylazine or isoflurane, and followed by cervical dislocation to assure death. This method is consistent with AVMA guidelines. We have followed all guidelines as rigorously set by Yale IACUC.

All human blood studies were approved by the Yale Human Investigation Committee (protocol# 1005006865). Informed consent was obtained from each subject and conform to the principles set out in the WMA Declaration of Helsinki and the Belmont Report. These are requirements for the Yale HIC. All data and sample use were specifically consented for by each subject. No studies were performed outside of what was consented for.

### Statistics

All data were expressed as mean $\pm$ SD. The parametric *t*-test was performed for comparisons of two groups. Comparisons of the group means were made with a Student's *t*-test or one-way ANOVA with a Bonferroni *post hoc* test (StatView V5.0, SAS). Analysis was performed with Prism software (GraphPad Software, Inc, La Jolla, CA). A difference of $P < 0.05$ was considered significant.

**Expanded View** for this article is available online.

### Acknowledgements

We thank Prof Xinran Liu (Director, Yale Biological EM Facility) for his helpful advice. We also thank Professor Silvio Inzucchi (Director, Yale Diabetes Center) for his assistance in diabetic patient recruitment. This study was supported by NIH-NHLBI grants RO1-122815, RO1-HL115247, and U54-HL117798 to J.H.

### Author contributions

SHL designed and performed most of experiments. WHT, JD, YX, DW, YJ, and AO performed some experiment. SHL, WHT, SL, KAM, and JH helped analyze the data. SHL, WHT, and JH, wrote and edited the manuscript. JS, KLL, SL, and KAM, edited the manuscript. JD, JS, GA, and GS recruited patients. AS and PM provided the Parkin$^{-/-}$ and PINK1$^{-/-}$ mice.

### Conflict of interest

The authors declare that they have no conflict of interest.

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
