## [Review Process File · EMBO Molecular Medicine]

Inducing mitophagy in diabetic platelets protects against severe oxidative stress

Seung Hee Lee, Jing Du, Jeremiah Stitham, Gourg Atteya, Suho Lee, Yaozu Xiang, Dandan Wang, Yu Jin, Kristen L. Leslie, GERALYN Spollett, Anup Srivastava, Praveen Mannam, Allison Ostriker, Kathleen A. Martin, Wai Ho Tang, John Hwa

Corresponding authors: John Hwa and Wai Ho Tang, Yale Cardiovascular Research Center, Yale University School of Medicine

Review timeline:

Submission date:	10 November 2015
Editorial Decision:	06 December 2016
Revision received:	10 March 2016
Editorial Decision:	04 April 2016
Revision received:	15 April 2016
Accepted:	18 April 2016

Transaction Report:

Editor: Céline Carret

1st Editorial Decision

06 December 2016

Thank you for the submission of your manuscript to EMBO Molecular Medicine. We have now heard back from the two referees whom we asked to evaluate your manuscript. Although the referees find the study to be of potential interest, they also raise a number of concerns that need to be addressed in the next final version of your article.

You will see that both referees are rather positive about the study while still highlighting some shortcomings that if satisfactorily addressed would improve the conclusiveness of the findings and provide additional understanding of the process. In addition, both referees spotted some text inconsistencies, and these should be corrected.

Given the balance of these evaluations, we feel that we can consider a revision of your manuscript if you can address the issues that have been raised within the 3-months constraints of a revision time. Please note that it is EMBO Molecular Medicine policy to allow only a single round of revision and that, as acceptance or rejection of the manuscript will depend on another round of review, your responses should be as complete as possible.

I look forward to receiving your revised manuscript.

***** Reviewer's comments *****

Referee #1 (Comments on Novelty/Model System):

The authors utilize human subjects as well as a mouse model. These models are adequate, although there are some unanswered questions regarding the phenotype of the mouse model, which I raise in my review.

Referee #1 (Remarks):

The manuscript by Lee et al, describes a series of studies in platelets isolated from human subjects, demonstrating a higher basal level of both ROS and autophagy/mitophagy in diabetic compared to healthy individuals. The authors provide substantial evidence for increased levels of mitochondrial damage in DM (diabetic mellitus) platelets and convincing evidence for increased mitophagy in these cells. The data presented are consistent with the model proposed in which platelets have an autonomous program for activating autophagy which is mediated by increased JNK activation in response to ROS, to recycle damaged mitochondria and ameliorate ROS in the cell. Further corroborating evidence in support of this model is obtained in mice where ablation of key mitophagy genes in mice (Parkin and PINK1) results in dramatic decrease in the time to occlusion of blood vessels in response to chemical challenge. The authors provide substantial evidence for the induction of mitophagy (including EM, confocal microscopy, biochemical analyses of autophagy/mitophagy markers) and have included key experimental controls throughout the manuscript. The data presented in general support the conclusions; however, there are some issues with data interpretation and unanswered questions, especially in terms of the pathways leading to mitophagy in response to ROS in platelets, which detract from overall impact.

Major issues:

1. Although a large number of autophagic markers are analyzed in healthy and DM platelets, the key upstream initiator of autophagy (Atg1/Ulk1) is not studied. Phosphorylation of Ulk1 needs to be analyzed.
2. If, according to the authors' model and as explained in the text referring to Figure 7, platelets from DM patients have higher levels of ROS and mitophagy, why are there higher levels of p-p53 (as shown in Figure 1), since mitophagy should conceivably degrade the mt-localized p53 ?
3. In experiments attempting to link increased ROS with mitophagy and p-JNK, the authors use normal (healthy) platelets, treat them with H₂O₂ and then reverse the observed changes with NAC. However, they also need to demonstrate that in DM platelets which have elevated ROS, treatment with NAC reverses the observed increases in these ROS markers.
4. The best readout for increased mitophagy is a decrease in overall mitochondrial DNA content. This assay needs to be incorporated in the analysis of both human and mouse platelets.
5. 3MA is a rather crude inhibitor of autophagy and has toxic effects in many cells. Additional, more specific inhibitors such as Spautin-1 or chloroquine (a fusion inhibitor) should be used in at least a few of these studies.
6. A key question from the mouse data is why do the platelets from the PINK^{-/-} mice (and presumably also from the Parkin^{-/-} mice) show such a dramatic propensity to aggregate and cause vessel occlusion in such a short time (1-2 minutes) following chemical challenge. Some additional biochemical data is needed to demonstrate which property of these platelets is compromised. If these platelets are more apoptotically sensitive (compared to those from WT animals) upon FeCl₂ challenge, this needs to be demonstrated. Moreover, can this sensitivity be reversed by treating mice with NAC (which can be administered in vivo)?

Minor points:

1. The results section can be shortened by moving some portions to the discussion, which is rather

short. For instance, there is a whole introductory paragraph in the results section before Fig.1 which is not necessary.

2. In the immunoblots showing patient samples, a legend is needed indicating the patient number (or code).
3. In Fig. 1E, the different numbers in each quadrant of the flow image is too small to discern.
4. There's a reference in the results section (top pg. 7) which is in a number format (instead of author name format).
5. The sentence "As a control, and CCCP is recognized...." on pg. 11 does not make sense, needs reformatting.

Referee #2 (Comments on Novelty/Model System):

This paper provides convincing data, authors have performed substantial work of quality, from observations in patients to molecular mechanistic insights with also the use of murine models, thus deserving publication in EMBO Molecular Medicine. However, some important drawbacks that I raised should be addressed for improvement before publication. Therefore, I recommend acceptance following revision.

Referee #2 (Remarks):

This paper by Lee and coll. entitled "The induction of mitophagy in diabetic platelets is protective against severe oxidative stress" investigates the importance of mitophagy in removal of damaged mitochondria in platelets in diabetic conditions (high glycemia) to prevent impaired thrombosis. This issue is crucial from a clinical point of view, and authors provide convincing evidence for underlying mechanisms. However, several points should be addressed.

- The most important point in my opinion is that authors claim that elevated ROS during DM is the cause of mitochondrial dysfunction promoting mitophagy, but they do not address the possibility that ROS increase in DM platelets could be the consequence of mitochondrial dysfunction. Could high glucose rate directly affect mitochondria?
- The paragraph at the beginning of the Results section is more a summary than an introductory paragraph thus useless.
- Results Paragraph 1 and Figure 1. The protein p53 is activated through post-translational modifications including phosphorylation on different sites: why authors do test only phosphorylation of serine 15? Fig.1I: show also % of healthy and damaged mitochondria in the HC cohort. Fig. S1: for HC n=4 in text but n= 5 in the legend of Fig.S1. Mitotracker: please precise which probe was used because some are not washed out of the cells when the mitochondrion's membrane potential is lost. Page 6: precise what is EM. ROS level is measured using dyes (which ones?) from Invitrogen, but kit from Enzo is mentioned in Methods section: please indicate precisely what was used.
- Results Paragraph 2 and Figure 2. Fig.2A&B: assess also BNIP3 and Nix in WB blots experiments. Fig.2C&D: is there autophagic vacuoles in HC?
- Results Paragraph 3 and Figure 3. The mTOR pathway actually inhibits autophagy, thus to completely get rid of its impact in platelets, other molecules should be evaluated such as S6K (reduction of its phosphorylation is expected) or use Rapamycin as autophagy inducer for example in the context of experiments shown in Figure 4.
- Results Paragraph 5 and Figures 5 and 6. The transition towards murine model is poor. Why experiments done with PINK1 KO mice (Figure 6) are different from experiments shown in Figure 5 using PARKIN KO mice? Also the experiments shown in Figure 6 are specific to the platelets research field thus not enough explained for a broad audience. What are PINK1 -/- DM mice?
- The manuscript is overall well written. However, some inaccuracies in the text bother somewhere the reading. In the reference to the Figures: page 7 bottom "(Supplementary Figure 3D)" instead of "(Supplementary Figure 3A)", page 11 "(Figure 5C-5D)" instead of "(Figure 4C-4D)". Page 9 "...3 fold change in H2O2 vs. HC..." instead of "...3 fold change in DM vs. HC ...". "the amount of energy required for mitophagy into the platelet is considerable" "prepackaged in the absence of a nucleus in platelets" the authors point to these notions at several locations in the Results section text (twice in Page 10) which is redundant and somehow useless.

- References: some references are incomplete, or in double (Kaneto 2010), or cited in the text as a number ("11" page 7), or without the year, or page 6 reference Michelle L. Boland 2013 not in the References list.... Please be more careful.

1st Revision - authors' response

10 March 2016

Response to Reviewer's

We thank the Reviewer's and the Editor for giving us the opportunity to revise our manuscript. We are grateful for the insightful and helpful comments provided by the Reviewer's that have significantly strengthened the manuscript. In response to the Reviewer's concerns we have reorganized the manuscript to improve clarity. Of particular importance, we performed many additional experiments in response to the Reviewer's concerns. The changes have been highlighted in the manuscript.

Referee #1:

We thank the Reviewer for the excellent comments and suggestions. We now provide additional experiments as requested by the Reviewer.

1. Although a large number of autophagic markers are analyzed in healthy and DM platelets, the key upstream initiator of autophagy (Atg1/Ulk1) is not studied. Phosphorylation of Ulk1 needs to be analyzed.

We thank the Reviewer for this excellent suggestion. Western blot analysis using Ulk and pUlk1(Ser757) antibodies demonstrated no significant change in diabetic human platelets (Appendix Figure Supplement 6A). Previous studies have suggested that phosphorylation of Ulk at Ser757 is induced by mTOR (Russell, Tian et al. 2013). Our results support that activation of mTOR is unchanged in DM platelets. As further support downstream targets of mTOR, p70S6K and S6 were also not activated in DM platelets (Appendix Figure Supplement 6B). These additional studies provide support that mTOR is not significantly involved in this DM platelet induced mitophagy pathway.

2. If, according to the authors' model and as explained in the text referring to Figure 7, platelets from DM patients have higher levels of ROS and mitophagy, why are there higher levels of p-p53 (as shown in Figure 1), since mitophagy should conceivably degrade the mt-localized p53 ?

The Reviewer highlights a very insightful and important concern. We were also initially perplexed by the high pp53 in the DM patients and mice in spite of the active mitophagy process. We believe that the key is acute versus chronic oxidative stress. Acute oxidative stress in normal platelets can be substantially reversed by mitophagy whereas DM platelets undergo continuing substantial stress arising from poorly controlled sugars, lipids, e.t.c. Thus mitophagy is only able to degrade and protect from pp53 to a degree, due to this high continuing oxidative stress associated with DM. In support of this, the absence of mitophagy (chemical or genetic inhibition) in DM, as also demonstrated with our mitophagy knockout studies, demonstrated further elevations of pp53 (Figure 4C-4F and Figure 6A). Many of the DM patients are very poorly controlled thus in spite of enhanced mitophagy there is still substantial pp53 and apoptosis occurring.

3. In experiments attempting to link increased ROS with mitophagy and p-JNK, the authors use normal (healthy) platelets, treat them with H₂O₂ and then reverse the observed changes with NAC. However, they also need to demonstrate that in DM platelets which have elevated ROS, treatment with NAC reverses the observed increases in these ROS markers.

In response to the Reviewer's comments we have now treated DM mice in vivo with NAC (15mg/kg intraperitoneally for 1 hour). This demonstrates a clear reduction in ROS and a reduction in platelet apoptosis (Appendix Figure Supplement 12A and 12B). The reduced ROS and JNK activation leads to no significant induction of mitophagy (Appendix Figure Supplement 12C). As now described, this demonstrates the key importance of high levels of ROS in the induction of the protective mitophagy process (Figure 7).

4. The best readout for increased mitophagy is a decrease in overall mitochondrial DNA content. This assay needs to be incorporated in the analysis of both human and mouse platelets.

This is an excellent suggestion from the Reviewer. To assess mitochondrial DNA content, we quantitated the human mitochondrial DNA for ND1 (NADH dehydrogenase subunit 1) and 16sRNA expression using specific primers in the human sample (Suliman, Carraway et al. 2007, N Dasgupta 2015) (Appendix Figure Supplement 5). We additionally quantified the mouse mitochondrial DNA for ND1 (Appendix Figure Supplement 5). All data was normalized to 18s RNA.

There were significant reductions with both human and mice platelet DM mitochondria contents when compared to healthy controls consistent with increased mitophagy.

5. 3MA is a rather crude inhibitor of autophagy and has toxic effects in many cells. Additional, more specific inhibitors such as Spautin-1 or chloroquine (a fusion inhibitor) should be used in at least a few of these studies.

This is an excellent suggestion from the Reviewer. Platelets from three DM patients were treated with 20 μ M Spautin-1 for 2h (Appendix Figure Supplement 9). Consistent with our previous results the inhibition of autophagy as observed with reduced LC3II resulted in increased pp53.

6. A key question from the mouse data is why do the platelets from the PINK1^{-/-} mice (and presumably also from the Parkin^{-/-} mice) show such a dramatic propensity to aggregate and cause vessel occlusion in such a short time (1-2 minutes) following chemical challenge. Some additional biochemical data is needed to demonstrate which property of these platelets is compromised. If these platelets are more apoptotically sensitive (compared to those from WT animals) upon FeCl₂ challenge, this needs to be demonstrated. Moreover, can this sensitivity be reversed by treating mice with NAC (which can be administered in vivo)?

We agree with the Reviewer and were also surprised with the dramatic nature of the vessel occlusion. It is recognized that apoptotic platelets induce clotting 50 to 100 times faster, the presence of phosphatidylserine on their surface (as detected by annexin V) acts as a catalytic site for clotting enzymes assembly and thrombin generation (Sinauridze, Kireev et al. 2007, Lannan, Phipps et al. 2014). Thus apoptosis of platelets is key to enhanced thrombosis associated with DM in the absence of mitophagy. As outlined by the Reviewer it is important to demonstrate whether these platelets from the knockout mice (mitophagy deficient) are more apoptotically sensitive and can potentially be reversed with NAC. When challenged with H₂O₂ (0.5 mM) the PINK1^{-/-} mice show markedly enhanced annexin V staining (Appendix Figure Supplement 11C). This increased apoptosis sensitivity is also observed in the Parkin^{-/-} mouse (Figure 5I). To confirm the increased sensitivity to apoptosis we have repeated these experiments using H₂O₂ 0.5mM and 1mM with subsequent significant increases in annexin V staining (Response Figure 1).

Response Figure 1. Absence of mitophagy during oxidative stress increases mitochondrial damage and apoptosis.

Significantly decreased mitochondria membrane potential and increased apoptosis in Parkin^{-/-} and PINK1^{-/-} mice platelets treated with H₂O₂.

As suggested by the Reviewer we also treated PINK1^{-/-} DM mice in vivo with NAC (intraperitoneal injection of 150mg/kg NAC for 1hr). NAC decreased ROS levels and decreased the % of Annexin-V positive platelets (Response Figure 2)

Response Figure 2. In vivo injection of NAC in PINK1-/- DM mice. Reduced ROS and apoptosis is observed with treatment of PINK1-/-DM mice with NAC.

Platelet ROS levels are decreased by NAC injection in PINK1-/- DM mice leading to decreased apoptotic platelets (annexin V staining). This arises from the reduced oxidative stress providing a similar result to **Appendix Figure Supplement 12**. As described in **Figure 7**, ROS plays a key role in triggering the protective mitophagy response. By reducing ROS, mitophagy is no longer induced (**Appendix Figure Supplement 12C**).

Minor points:

1. The results section can be shortened by moving some portions to the discussion, which is rather short. For instance, there is a whole introductory paragraph in the results section before Fig.1 which is not necessary.

Thank you. We have shortened the results section as suggested.

2. In the immunoblots showing patient samples, a legend is needed indicating the patient number (or code).

Patient numbers according to when they were recruited have now been added as suggested.

3. In Fig. 1E, the different numbers in each quadrant of the flow image is too small to discern.

We have now presented a larger flow image size

4. There's a reference in the results section (top pg. 7) which is in a number format (instead of author name format).

Thank you. It has been changed

5. The sentence "As a control, and CCCP is recognized...." on pg. 11 does not make sense, needs reformatting.

Thank you. We have changed the sentence.

Referee #2:

We appreciate the very insightful comments presented by the Reviewer. In response to the concerns we have performed additional experiments and substantially reorganized the manuscript.

1. The most important point in my opinion is that authors claim that elevated ROS during DM is the cause of mitochondrial dysfunction promoting mitophagy, but they do not address the possibility that ROS increase in DM platelets could be the consequence of mitochondrial dysfunction. Could high glucose rate directly affect mitochondria?

This is an excellent point raised by the Reviewer. We have previously demonstrated that glucose leads to increased ROS and mitochondrial dysfunction through an aldose reductase dependent pathway (Tang, Stitham et al. 2014) (Response Figure 3). What we have been most excited about in this present study is the presence of a robust mitophagy pathway in the short lived and anuclear platelet that serves to protect against such ROS inducing glucose stressors.

Annexin V

Response Figure 3.

Treatment of platelets with increasing glucose concentrations demonstrates progressive decreased mitochondrial function and increased apoptosis (adapted from Tang, Stitham et al 2014).

2. The paragraph at the beginning of the Results section is more a summary than an introductory paragraph thus useless.

Thank you for this suggestion. We have removed the paragraph as suggested.

3. Results Paragraph 1 and Figure 1. The protein p53 is activated through post-translational modifications including phosphorylation on different sites: why authors do test only phosphorylation of serine 15?

The Reviewer is correct in pointing out that many sites can be phosphorylated on p53. We have performed extensive phosphoproteomic arrays in assessing the DM and apoptotic platelets and confirmed that Ser15 is the key phosphorylation site in platelets leading to mitochondrial dysfunction and apoptosis. No significant changes were observed with Ser46 and Ser 392.

4. Fig.1I: show also % of healthy and damaged mitochondria in the HC cohort.

Thank you. We have now added % of healthy and damaged mitochondria in the HC in Figure1J. Approximately 85% of mitochondria are healthy and 14 % are unhealthy in HC platelets, likely reflecting basal mitochondrial turnover.

5. Fig. S1: for HC n=4 in text but n= 5 in the legend of Fig.S1.

We apologize for this error. We used 4 HC platelets for these experiments. The error has been corrected.

6. Mitotracker: please precise which probe was used because some are not washed out of the cells when the mitochondrion's membrane potential is lost.

As suggested by the Reviewer the precise probes have now been added to the manuscript. Washed platelets (1×10^6 cells/ml) were allowed to settle on glass-bottom dishes for 1h and stained mitochondria using Mitotracker (Lifescience, MitoTracker Red M7512, or deep Red FM, M22426) for 15 min prior to fixing with 4% paraformaldehyde solution (Santa cruz Biotechnology).

7. Page 6: precise what is EM.

Thanks. EM means Electron Microscopy.

8. ROS level is measured using dyes (which ones?) from Invitrogen, but kit from Enzo is mentioned in Methods section: please indicate precisely what was used.

We used a number of different assays (most suitable to the experimental samples) to confirm and validate our results. We used CellROX green (Molecular Probes, C10444) for Figure1A experiments, and we used ROS detection kit from Enzo for the experiments in Figure 3C. A new section has been added to Methods describing the assays used.

9. Results Paragraph 2 and Figure 2. Fig.2A&B: assess also BNIP3 and Nix in WB blots experiments.

We thank the Reviewer for this suggestion. We have recruited additional DM patients and assessed for BNIP3L/NIX (Appendix Figure Supplement 4). There are increases in BNIP3L/NIX in these platelets supporting the presence of an enhanced mitophagy process.

10. Fig.2C&D: is there autophagic vacuoles in HC?

There are the occasional autophagy vacuoles in HC platelets however very few were detected compared to the DM platelets. This is reflected by the occasional LC3 staining even in HC which represents a degree of basal autophagy.

11. Results Paragraph 3 and Figure 3. The mTOR pathway actually inhibits autophagy, thus to completely get rid of its impact in platelets, other molecules should be evaluated such as S6K (reduction of its phosphorylation is expected) or use Rapamycin as autophagy inducer for example in the context of experiments shown in Figure 4.

Thank you for these excellent suggestions. We have now assessed pp70S6K and pS6 in DM platelets (Appendix Figure Supplement 6B). Consistent with the absence of significant changes in mTOR there were no significant changes, supporting that mTOR likely does not play an important role in the DM induced mitophagy process.

12. Results Paragraph 5 and Figures 5 and 6. The transition towards murine model is poor. Why experiments done with PINK1 KO mice (Figure 6) are different from experiments shown in Figure 5 using PARKIN KO mice? Also the experiments shown in Figure 6 are specific to the platelets research field thus not enough explained for a broad audience. What are PINK1 -/- DM mice?

We apologize for the lack of clarity and hope that the revised manuscript is much improved. Both the PINK1 and Parkin knockout mice represent two different mouse lines where mitophagy has been knocked out. Rather than repeating everything in exactly the same fashion (and presenting redundant data) we wanted to present complementary data which support the importance of mitophagy in DM platelets using different techniques. In response to the Reviewer's concerns we have now repeated some of the Parkin-/- experiments in the PINK1-/- mice and demonstrated that there is lack of induction of mitophagy upon increased oxidative stress associated with increased platelet apoptosis (Appendix Figure Supplement 11). Such data was consistent with, and is supported by, the human data.

13. The manuscript is overall well written. However, some inaccuracies in the text bother somewhere the reading. In the reference to the Figures: page 7 bottom "(Supplementary Figure 3D)" instead of "(Supplementary Figure 3A)", page 11 "(Figure 5C-5D)" instead of "(Figure 4C-4D)". Page 9 "...3 fold change in H₂O₂ vs. HC..." instead of "...3 fold change in DM vs. HC ...". "the amount of energy required for mitophagy into the platelet is considerable" "prepackaged in the absence of a nucleus in platelets" the authors point to these notions at several locations in the Results section text (twice in Page 10) which is redundant and somehow useless.

We thank the Reviewer for the comment that the manuscript is "overall well written". We apologize for the inconsistencies and redundancies, and have corrected them as suggested.

14. References: some references are incomplete, or in double (Kaneto 2010), or cited in the text as a number ("11" page 7), or without the year, or page 6 reference Michelle L. Boland 2013 not in the References list.... Please be more careful.

We apologize for these errors. They have been corrected.

Thank you for the submission of your revised manuscript to EMBO Molecular Medicine. We have now received the enclosed reports from the referees that were asked to re-assess it. As you will see the reviewers are now supportive and I am pleased to inform you that we will be able to accept your manuscript pending editorial final amendments.

1) Please address the minor text changes commented by referee 2. We would like to encourage you to also add the figures to referee 1 points 1 and 2 as Appendix figures. Please provide a letter INCLUDING the reviewer's reports and your detailed responses to their comments (as Word file).

Please submit your revised manuscript within two weeks. I look forward to seeing a revised form of your manuscript as soon as possible.

***** Reviewer's comments *****

Referee #1 (Comments on Novelty/Model System):

I believe that on technical grounds as well as innovation, this is a high-caliber manuscript. The medical impact is not readily apparent; however, it lays the foundations for potential therapeutic intervention in the future if agents which increase basal autophagy levels are developed.

Referee #1 (Remarks):

The manuscript by Lee et al has been substantially improved. Additional data confirming mitophagy are presented. Moreover, a mechanistic explanation supported by additional data is provided for the rapid induction of aggregation in autophagy/mitophagy-deficient platelets (the data presented in "Response Figures 1 and 2" should be included in the manuscript, even as supplemental data). Overall, the work described in this manuscript provides a novel and potentially clinically significant mechanism for the role of mitophagy in aggregation of platelets in diabetic conditions.

Referee #2 (Remarks):

The authors have satisfactorily addressed the concerns that I raised, and greatly improved the publication.

Minor suggestions to further ameliorate:

Figure 1. MitoTracker probe can also be used to assess mitochondria number, which is decreased in DM platelets as shown by EM (Fig.1I). Thus reduced Mitotracker fluorescence in DM platelets (Fig.1D) could also result from reduced mitochondria number, additionally (or not) to MMP decrease. Authors should discuss this point.

Page 9 and Figure 3C: "High glucose alone induced only a small increase in ROS (1.5 fold change)": it seems less than 1.5 on Figure 3C.

This point is also important regarding the first issue that I raised. HG seems to directly induce mitochondrial damage and not ROS increase which is thus likely the consequence of mitochondrial stress.

Page 13 "When stressed with DM levels of oxidative stress...": this part of the sentence is weird.

Page 13: DM mice: it is not clear what are these DM mice in both Results and Methods sections.

Figure S4B: error: "Bnip2" instead of "Bnip3L"

Figure S12B: error: "unstaining" not shown

2nd Revision - authors' response

15 April 2016

Referee #1:

We thank the Referee for the comments regarding our manuscript being a "high-caliber manuscript".

The manuscript by Lee et al has been substantially improved. Additional data confirming mitophagy are presented. Moreover, a mechanistic explanation supported by additional data is provided for the rapid induction of aggregation in autophagy/mitophagy-deficient platelets

(the data presented in "Response Figures 1 and 2" should be included in the manuscript, even as supplemental data).

As requested, we have now moved Response Figure 1 and 2 to the Appendix Figure S11D/E and Appendix Figure S12D/E, respectively.

Referee #1:

We thank the Referee for the comment that the manuscript is greatly improved. We now address the remaining minor suggestions.

1. Figure 1. MitoTracker probe can also be used to assess mitochondria number, which is decreased in DM platelets as shown by EM (Fig.1I). Thus reduced Mitotracker fluorescence in DM platelets (Fig.1D) could also result from reduced mitochondria number, additionally (or not) to MMP decrease. Authors should discuss this point.

That is an excellent point raised by the Referee regarding the reduced MitoTracker probe reflecting reduced mitochondrial numbers. This has now been added to the Results as suggested.

2. Page 9 and Figure 3C: "High glucose alone induced only a small increase in ROS (1.5 fold change)": it seems less than 1.5 on Figure 3C.

This point is also important regarding the first issue that I raised. HG seems to directly induce mitochondrial damage and not ROS increase which is thus likely the consequence of mitochondrial stress.

We thank the Referee for highlighting this important point. High glucose application to normal healthy platelets induces mitochondrial stress with a small increase in ROS. There is no substantial mitochondrial damage in comparison to the exposure to severe oxidative stressors.

3. Page 13 "When stressed with DM levels of oxidative stress...": this part of the sentence is weird.

Thanks. This has now been clarified.

4. Page 13: DM mice: it is not clear what are these DM mice in both Results and Methods sections.

The DM mice were produced using 5 days STZ (50mg/kg in each day) and 12 weeks high fat diet. This has now been highlighted in Material and Methods.

5. Figure S4B: error: "Bnip2" instead of "Bnip3L"

Thanks. The error has been corrected.

6. Figure S12B: error: "unstaining" not shown

Thanks. The error has been corrected.

Corresponding Author Name: John Hwa
 Journal Submitted to: EMBO Molecular Medicine
 Manuscript Number: EMM-2015-06046